# A rotary mechanism for allostery in bacterial hybrid malic enzymes

Christopher John Harding [1✉], Ian Thomas Cadby [1], Patrick Joseph Moynihan [1] & Andrew Lee Lovering [1✉]

Bacterial hybrid malic enzymes (MaeB grouping, multidomain) catalyse the transformation of malate to pyruvate, and are a major contributor to cellular reducing power and carbon flux. Distinct from other malic enzyme subtypes, the hybrid enzymes are regulated by acetyl-CoA, a molecular indicator of the metabolic state of the cell. Here we solve the structure of a MaeB protein, which reveals hybrid enzymes use the appended phosphotransacetylase (PTA) domain to form a hexameric sensor that communicates acetyl-CoA occupancy to the malic enzyme active site, 60 Å away. We demonstrate that allostery is governed by a large-scale rearrangement that rotates the catalytic subunits 70° between the two states, identifying MaeB as a new model enzyme for the study of ligand-induced conformational change. Our work provides the mechanistic basis for metabolic control of hybrid malic enzymes, and identifies inhibition-insensitive variants that may find utility in synthetic biology.

[1] Department of Biosciences, University of Birmingham, Birmingham, UK. ✉email: cjh30@st-andrews.ac.uk; a.lovering@bham.ac.uk

Malic enzymes (MEs) catalyse the oxidative decarboxylation of L-malate to produce pyruvate and $CO_2$, coupled to the reduction of an $NAD(P)^+$ cofactor. MEs are widely-distributed enzymes found in almost all organisms, from bacteria to higher eukaryotes. Moreover, multiple MEs isoforms have been identified, which are categorised into three groups based on cofactor preference and the ability to decarboxylate oxaloacetate (EC 1.1.1.38-40). The placement of MEs at the key phosphoenolpyruvate-pyruvate-oxaloacetate node (Fig. 1A) provides an influence over major carbon metabolism routes, linking glycolysis to the TCA cycle[1]. In general, $NAD^+$ dependent MEs function to provide pyruvate for the TCA cycle, whereas $NADP^+$ dependent MEs function to generate NADPH for anabolic reactions. Diverse roles of MEs include: energy metabolism; the biosynthesis of fatty acids and steroids[2,3]; metabolism of glutamine as an alternative energy source in rapidly proliferating cells such as tumours[4–6]; providing $CO_2$ for fixation by RuBisCO in photosynthesis[7,8] and in plant defences and oxidative stress responses[9].

A wide range of ME isoforms have been described, differing physically and enzymatically. Eukaryotic ME isoforms (cytosolic, mitochondrial and plastidic) have been well characterised, typically forming homotetramers from 60 kDa subunits[10–14]. Conversely, bacterial ME isoforms are comparatively understudied and collectively demonstrate greater structural and functional diversity (ranging from "minimal" 40 kDa subunits to much larger 85 kDa multidomain proteins)[15–20]. The greater bacterial ME complexity arises from a need for allosteric regulation due to the non-compartmentalisation of the bacterial cell.

The larger ME isoforms (hybrid enzymes) are predicted to be a fusion of two unrelated enzymes; a catalytic N-terminal ME subunit and a C-terminal phosphotransacetylase (PTA) domain. The PTA domain is homologous to PTA enzymes (that interconvert acetyl-CoA and phosphate to CoA and acetyl phosphate), yet they appear to lack critical catalytic residues and thus acts as a

sensor instead[21]. Hybrid MEs are widespread throughout bacteria, the archetypes being DME and TME from *Sinorhizobium meliloti*[22] and MaeB from *Escherichia coli*[23]. Partial kinetic characterisations of these enzymes have facilitated suggestions of their metabolic roles and suggested that the PTA domain could be important for allosteric regulation.

DME, an $NAD(P)^+$ hybrid enzyme, has been reported to play an important role in symbiotic $N_2$ fixation[24,25]. The homologous TME is suggested to function instead as an NADPH and acetyl-CoA generator (required for anabolic reactions, the latter made indirectly), similar to other $NADP^+$ specific enzymes like *E. coli* MaeB[23]. These products provide a carbon flux to phosphoenolpyruvate, for gluconeogenesis. Hence, MaeB was shown to be required for normal growth on TCA cycle intermediates or acetate. Multiple metabolic effectors, most notably acetyl-CoA, regulate MaeB activity. A C-terminal truncation of MaeB was unable to oligomerise and became insensitive to acetyl-CoA inhibition, suggesting that allosteric regulation is mediated by the PTA domain[23].

Without structural information on hybrid enzymes, the exact nature of ME:PTA coupling and allosteric regulation has remained unclear. Here we demonstrate that a representative hybrid ME from the predatory bacterium, *Bdellovibrio bacteriovorus* (Bd1833/MaeB, chosen because of the importance of metabolic efficiency to obligate predators), forms a large complex whose assembly is centred around a PTA regulatory hexamer. From structures of both bound and active states of the isolated domains and full-length enzyme, we reveal that acetyl-CoA, a potent allosteric inhibitor of ME activity, interacts with a conserved pocket of the PTA domain, driving conformational changes that are propagated 60 Å to the ME active site. These conformational changes are reliant upon several conserved structural adaptations that are unique to hybrid enzymes. Transition between the inhibited and active states requires a large-scale ~70° rotation of the enzymatic domains relative to the PTA

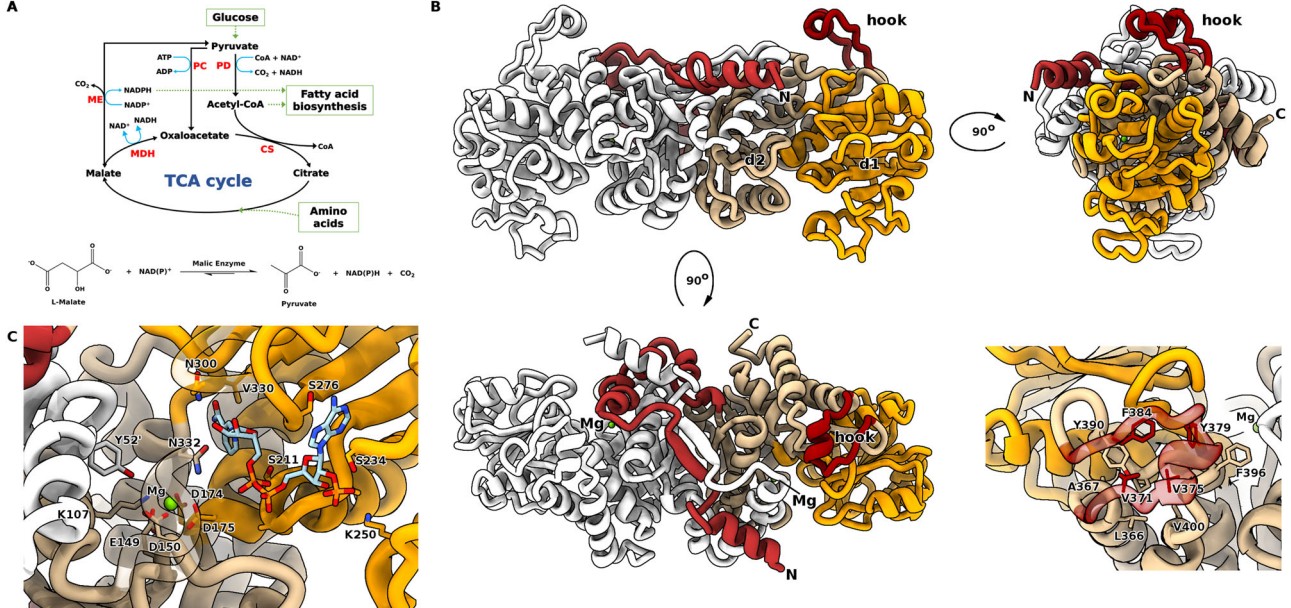

**Fig. 1 Function and structural features of the malic enzyme domain. A** Overview of MaeB malic enzyme (ME) metabolic context, with malate dehydrogenase (MDH), pyruvate carboxylase (PC), pyruvate dehydrogenase (PD) and citrate synthase (CS); reaction outlined beneath. The ME:MDH:PC cycle is capable of acting as a transhydrogenase. **B** Structure of the MaeB_ME domain dimer, shown from three orthogonal views, with crossover (red–brown), d1 (orange), d2 (beige) and hook (red) subdomains. Lower righthand panel illustrates hydrophobic packing between the hook and the d1:d2 interface. **C** Interactions observed in the MaeB_ME:NADP+ complex, relevant residues in contact with NADP+ (light blue) are shown in stick form. The loop over the Mg ion is rendered transparent for clarity; the crossover region donates Y52 (labelled with a prime) to the opposing active site.

oligomer, restoring a productive ME conformation. Together, our structural and kinetic analyses provide a mechanistic basis for hybrid ME allostery.

## Results

**Hybrid MEs use a classical catalytic fold with an appended "Hook" subdomain**. To understand the mechanistic basis for regulation of hybrid MEs, we first sought to determine the structures of the catalytic and regulatory domains in isolation. We solved the crystal structure, to 1.67 Å, of the first 434 amino acids of MaeB, corresponding to the N-terminal catalytic domain (MaeB$_{ME}$, Table S1). The structure was solved using molecular replacement with the minimal ME of *Candidatus* Phytoplasma (PDB: 5CEE) as a search model, with residues 15–425 traceable. Structurally, MaeB$_{ME}$ closely resembles the fold of small prokaryotic MEs (Fig. 1B), which form a tightly interacting dimer where the catalytic site is composed of residues from both protomers[26]. Dimerisation in solution was confirmed by size-exclusion chromatography (Fig. S1). MaeB$_{ME}$ can be divided into three subdomains: region d1 which contains a Rossmann-like fold that forms the nucleotide binding site; region d2 which is mixed α/β and contains an N-terminal extension that sits over the dimer interface and extends into the opposing monomer; and a small "hook-like" subdomain (residues 369–386) that has no structural equivalent in other characterised ME folds and is best-described as a projecting loop with central alpha helix. This nomenclature identifies d1 as the main functional domain of the enzyme, but means that d2 precedes it in the amino acid sequence. Comparatively, the hook region is easily discerned as an insert in the primary sequence of hybrid MEs (Fig. S2), and packs against d1 via several conserved hydrophobic residues. In comparison to the well-characterised eukaryotic MEs, d1 and d2 are equivalent to domains C and B, respectively, with MaeB omitting the tetramerization domain[11]. MaeB$_{ME}$ possesses all the consensus residues for catalysis, grouped around a Mg$^{2+}$ ion binding site situated at the d1:d2: hook interface (Fig. 1B, C), including the putative proton donor E149, D174 which coordinates the general acid/base K107, and Y52, which is responsible for the final tautomerisation step[27]. Each of these catalytic residues is in an equivalent pose to those from the eukaryotic enzymes (Fig. S3B), with the difference that Y52 is contributed from the other monomer rather than on the same chain.

MaeB$_{ME}$ possesses a putative classical nucleotide binding motif in d1 (207-GAGASA-212), and to understand the precise nature of cofactor:enzyme contacts, we obtained a structure in complex with NADP$^+$ at 1.67 Å resolution (Figs. 1C and S4). When compared to the unbound state, there is minimal rearrangement upon binding NADP$^+$. Specificity for NADP$^+$ is likely governed by direct interaction with K250, and the use of R242 to prevent phosphate group repulsion by neutralising D233 (akin to that observed for the pigeon liver enzyme 1GQ2)[13]. The nicotinamide group contacts V330 and N300, and is positioned <7 Å from the bound Mg$^{2+}$ cation, allowing adequate space for binding of substrate.

**Kinetic analysis of MaeB and its regulation by acetyl-CoA**. To determine the catalytic activity and specificity of MaeB, a series of kinetics experiments were carried out on MaeB and various mutant forms of MaeB (Fig. 2 and Table 2). The catalytic activity of MaeB (full-length protein) highly favours the forward malate oxidative decarboxylation reaction ($k_{cat}/K_M = 10.0$ s$^{-1}$mM$^{-1}$) in comparison to the reverse pyruvate reductive carboxylation reaction ($k_{cat}/K_M = 0.03$ s$^{-1}$mM$^{-1}$). MaeB activity required the presence of a divalent cation (Mn$^{2+}$ or Mg$^{2+}$) and the cofactor NADP$^+$ (no activity detected in the presence of NAD$^+$). The substrate concentration of L-malate was limited to 10 mM in ME activity assays, due to an observed reduction in rates when 10 mM L-malate was exceeded (Fig. S5C).

The activity of MaeB was assayed in the presence of a number of known ME activity regulators, (Fig. S5D)[23,28]. Oxaloacetate and acetyl-CoA were found to potently inhibit the activity of MaeB (<10% of uninhibited levels), in comparison to CoA and acetyl phosphate that moderately inhibited the activity (~50% of uninhibited levels), (Fig. 2A). Our truncated ME construct (MaeB$_{ME}$) retained activity, but was insensitive to inhibition by acetyl-CoA, CoA and acetyl phosphate (Fig. 2B and Table S2), confirming that these allosteric inhibitors act at the appended PTA domain. Contrastingly, oxaloacetate inhibited both MaeB and MaeB$_{ME}$ activities, suggesting this effector interacts directly with the ME domain.

The mode of inhibition by acetyl-CoA and CoA was probed by performing steady state kinetics experiments in the presence of several fixed inhibitor concentrations (with either fixed L-malate or NADP$^+$ substrates). Global fitting to different inhibition models suggested acetyl-CoA acts as a non-competitive inhibitor when L-malate was fixed ($V_{max}$ markedly decreased and the apparent $K_M$ remained unaffected), which is further supported graphically by a Lineweaver–Burk plot (Fig. S5I, J). From this data a $K_{i \text{ (acetyl-CoA)}}$ value of 0.70 μM was extracted. Unfortunately, steady state kinetic plots where NADP$^+$ was fixed, have a large degree of error, due to the restriction of available L-malate concentrations, leading to ambiguity of $V_{max}$ values at high inhibitor concentrations (Fig. S5E–H). As a consequence, this data was not used to extract kinetic parameters. Instead, a comparison of acetyl-CoA and CoA inhibitor potency was investigated by performing dose response assays (fixed L-malate and NADP$^+$ concentrations and varied inhibitor concentrations), (Fig. 2E, F). The IC$_{50}$ value obtained for acetyl-CoA (0.45 μM) is ~45 times smaller than the IC$_{50}$ determined for CoA (20.19 μM). This finding highlights that MaeB activity is far more sensitive to changes in the concentration of acetyl-CoA rather than CoA. We therefore sought to characterise the mechanism underpinning PTA-mediated regulation.

**The PTA regulatory domain adopts a hexameric ring structure with co-operatively arranged acetyl-CoA binding sites**. We generated an N-terminally truncated mutant, MaeB$_{PTA}$, spanning residues 439–780, and determined its structure to 1.96 Å resolution. The structure reveals a departure from characterised PTA proteins, forming a hexameric ring with D3 symmetry (Fig. 3A). The classical PTA dimer[29] is formed from one unit from the "front" trimer, and one unit from the "back" trimer, giving a triangular array with dimensions of ~120 Å across by 60 Å deep. The PTA fold can be divided into two subdomains—d1, composed of the N and C-terminal regions (residues 441–595 and 738–776), and d2 (residues 596 to 737). Together, the hexameric assembly unique to MaeB has two isologous interfaces (isologous defined as the same face of contact, d2:d2 at dimer interface halfway along the triangle edge, d1:d1 at triangular outer corner) and one complex heterologous interface (d1:d2 at the inner corner of the triangle). The peptide region that would link to the enzymatic ME domain in the full-length protein is situated at the PTA N-terminus, sat below the dimer interface. The d2:d2 dimer interface is largely similar to that observed in existing PTA proteins that are limited to a dimeric oligomerization, revealing that the MaeB$_{PTA}$ hexamer results from adaptations to the d1 region that allow shape complementarity at multiple points of the trimer interface (Fig. 3A, C, D). HPLC analyses confirm that this protein construct predominantly co-purifies with the allosteric

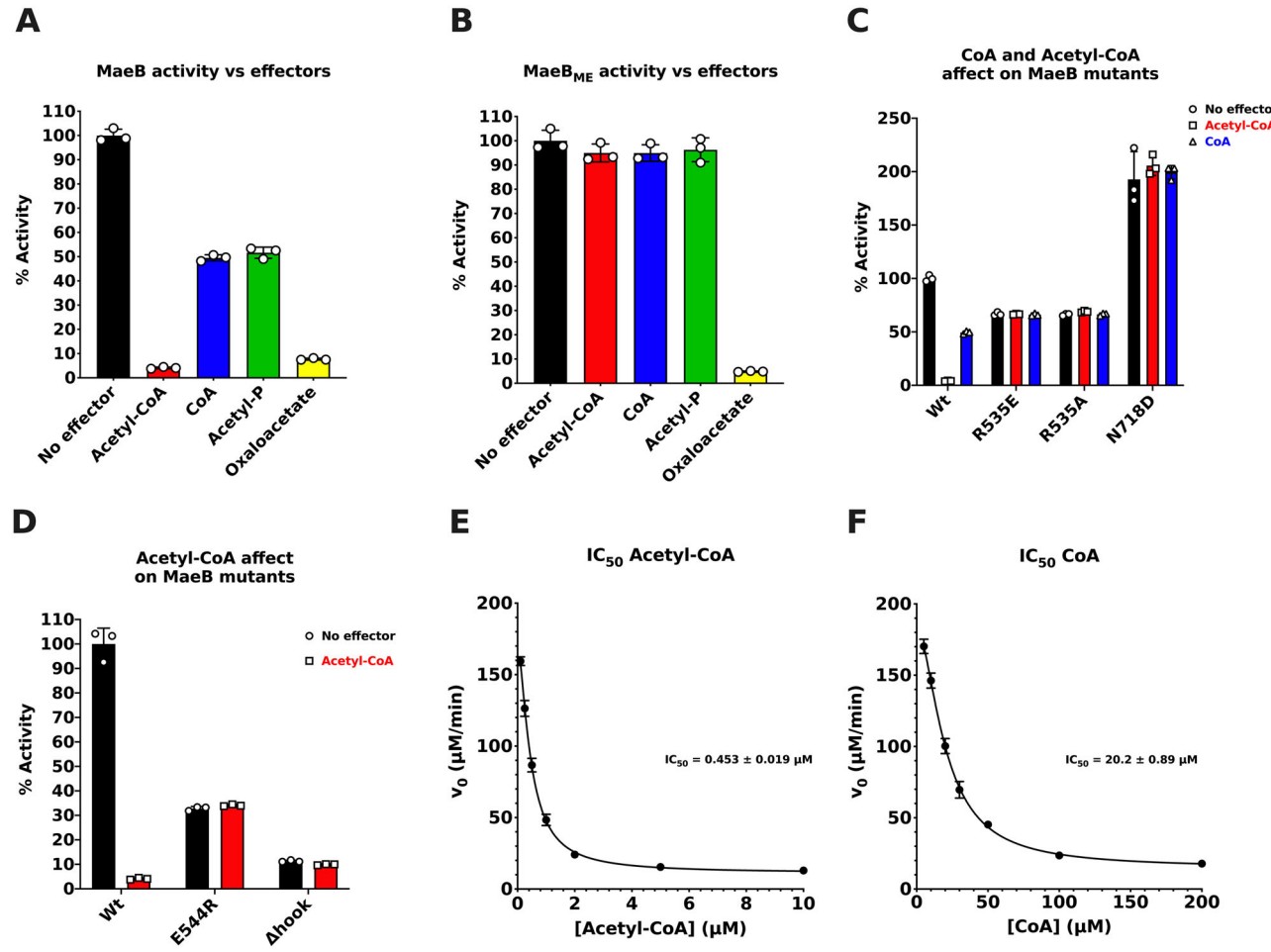

**Fig. 2 Enzyme kinetics data of MaeB and variant proteins. A** The % of MaeB activity in the presence of effectors in comparison to standard reaction conditions. MaeB is potently inhibited by oxaloacetate and acetyl-CoA. **B** The % of MaeB$_{ME}$ (ME domain only) activity in the presence of effectors in comparison to standard reaction conditions. MaeB$_{ME}$ is not inhibited by acetyl-CoA, CoA or acetyl phosphate, suggesting these molecules interact solely with the PTA domain. **C** The % activity of MaeB mutants, designed to distrupt the CoA binding pocket. R535A, R535E and N718D all became insensitive to inhibition by acetyl-CoA and CoA. R535 mutants have a reduced activity compared to wt, whereas the N718D mutant has an increased activity. **D** Comparison of the % activity of MaeB$_{\Delta hook}$ and MaeB-E544R in the presence of acetyl-CoA. Datasets are normalised to the activity of MaeB under normal reaction conditions. See materials and methods for concentrations of effectors used. (**E** + **F**) IC$_{50}$ plot for acetyl-CoA and CoA, respectively. Acetyl-CoA IC$_{50}$ value is ~45 times smaller than that of CoA, suggesting MaeB is far more sensitive to changes in acetyl-CoA concentration. The data is plotted showing the mean value from triplicate experiments, with standard deviation error bars. Source data are provided in the Source Data file.

regulatory ligand acetyl-CoA (Fig. S6A). In contrast to the difficulties encountered when mapping acetyl-CoA binding sites in catalytically active PTA enzymes[30,31], we were able to unambiguously model the mode of interaction (Figs. 3B and S4). Acetyl-CoA sits in a shallow groove formed between d1 and d2, placing the bulk of the ligand at the inner corner of the trimer. A pocket formed from Y576, T614, N718, Y721, K722, P735 and Q748 surrounds the acetate moiety; a single acetyl-binding residue, Q683, is provided by the opposing monomer of the dimer (Fig. S7). These residues exhibit some variation in homologues (Fig. S2). An extension from d1 (residues 524–550, forming a loop with two helices that we term the 3′ loop) sits over the 3′-phosphate group, interacting via residues R535 and K538. The central pyrophosphate region of acetyl-CoA sits flat against a region of d1 (labelled "face helix"), making an electrostatic interaction with R581. A loop from d2 interacts with the adenine ring; residue N616 hydrogen bonding to atom N6. On the d1:d1 isologous edge of the hexamer, two 3′ loops form an interface (Fig. 3D–F), presumably contributing to propagation/co-operativity of any induced conformational change upon binding or loss of ligand.

In agreement with studies on *E. coli* and *S. meliloti* MaeB, we did not observe any PTA enzymatic activity; this is also consistent with deviation from characterised PTA enzyme sequence consensus at positions V746 and Y576 (an aspartate and serine/threonine in active enzymes, respectively) (Fig. S3C).

**Structure-guided methods allow for design of variants insensitive to acetyl-CoA Inhibition.** Our structure of the acetyl-CoA bound form (MaeB$_{PTA}$), which licences inhibition of full-length MaeB, prompted us to investigate the means of allosteric signalling and the transition between active and inactive conformations triggered by acetyl-CoA binding to the PTA domain. Using HPLC as a readout (Fig. S6), we were able to prevent acetyl-CoA binding by mutating R535 that makes a direct contact to the 3′ phosphate group. Variants R535E and R535A abolished acetyl-CoA binding, and generated an enzyme insensitive to allosteric inhibition by both CoA and acetyl-CoA (Fig. 2C and Table S2), although these mutants retained ~70% activity to wild-type (wt) MaeB. As a control to mitigate any local/biased effects introduced by 3′ loop mutation, we designed a different unbound

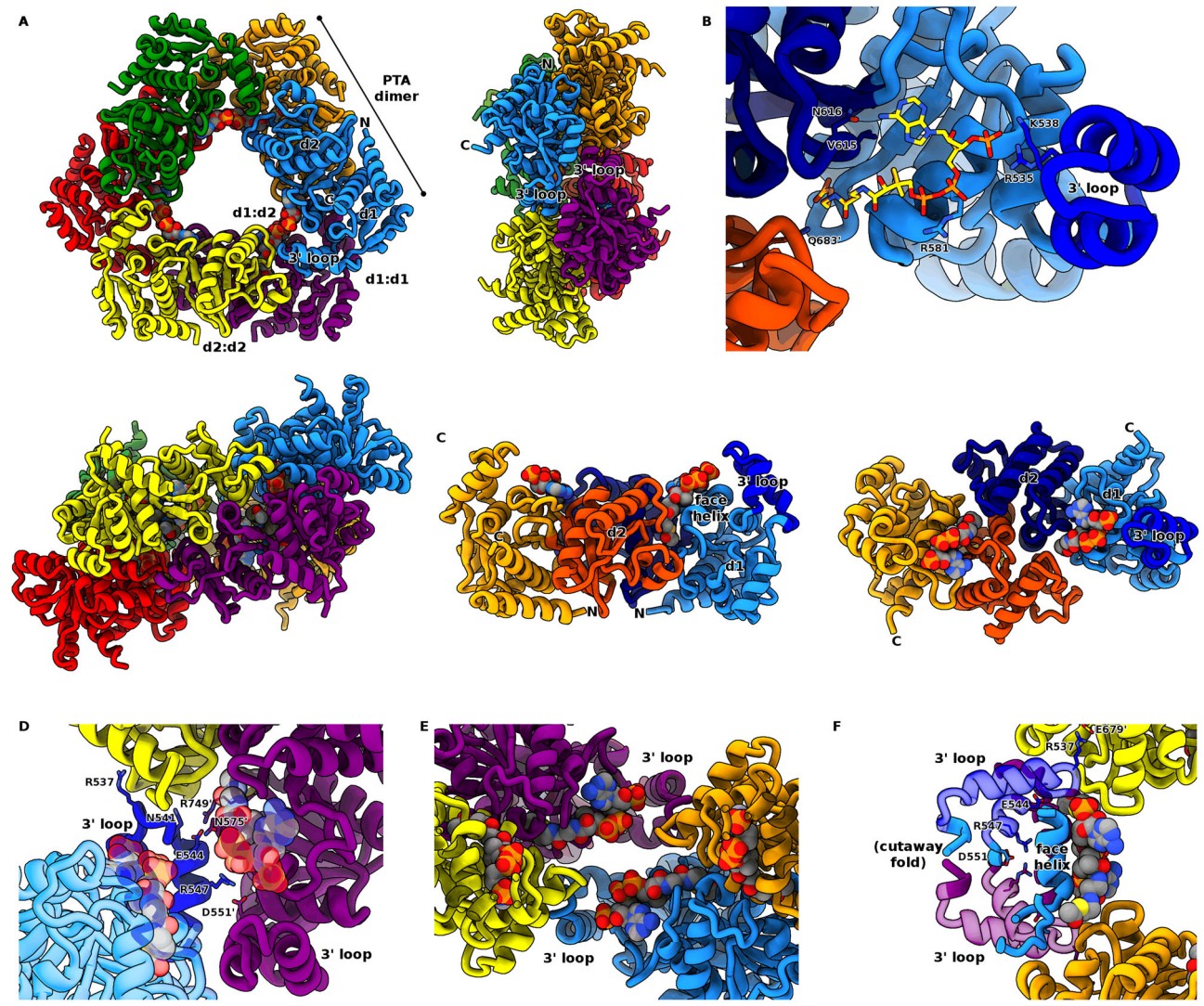

**Fig. 3 Structure of the MaeB PTA hexamer and interactions with acetyl-CoA. A** Structural features of the D3 MaeB$_{PTA}$ hexamer, viewed from front, side and edge orientations. The classical PTA dimer (e.g. orange and blue protomers) trimerizes to give d1:d1, d2:d2 and d1:d2 interfaces. The N-terminal connection to the (missing) ME domain is situated at the hexamer long edge. **B** Interaction of the MaeB$_{PTA}$ domain with the regulatory acetyl-CoA ligand (yellow). A helical motif, which we term the 3′ loop (R535 and K538), interacts at the ligand 3′ phosphate. **C** Features of the PTA dimer unit (from two orthogonal views), which associates around the d2:d2 interface, and binds two molecules of acetyl-CoA at the d1:d2 cleft. **D, E** Involvement of the 3′ loop in forming the trimer interface, acetyl-CoA shown in vdw form. Two 3′ loops form the inner "cradle" at the d1:d1 interface, such that two acetyl-CoA binding sites come into close contact between separate dimers. **F** Cutaway view of the d1 face helix forming a flat surface for acetyl-CoA interaction.

variant by mutation at the opposite end of the binding site—a N718D mutant repels ligand at the acetyl group. Supportively, N718D protein purifies without bound ligand and is also insensitive to inhibition (Fig. 2C and Table S2). Interestingly, the N718D variant had increased activity compared to wt MaeB, indicative of release from inhibitor suppression. To verify these findings without using mutation, we were inspired by flavoprotein methods used to remove tightly-bound FMN[32], and used a moderate Br$^−$/urea wash step to generate ligand-free MaeB. HPLC analyses confirmed we were able to generate an apoprotein via this strategy (Fig. S6B) and remarkably kinetics assays revealed that this process boosted activity of the full-length protein (like the N718D mutant), providing confirmation that renatured samples were catalytically competent and no longer suppressed by co-purified ligand (Table S2).

**Apoprotein structures reveal correlated motions between the inhibited and active PTA forms.** Washed samples of the native

MaeB$_{PTA}$ PTA-only construct (hereafter referred to as PTA$_{apo}$) were crystallised and diffracted to 2.72 Å resolution. We were also able to generate structures for R535E, R535A and N718D PTA domain variants, which demonstrate the same features as the wt PTA$_{apo}$ structure. PTA$_{apo}$ retains the same D3 symmetry as the acetyl CoA bound form, and, in agreement with our HPLC data (Fig. S6C), there was a clear lack of ligand electron density. Loss of ligand binding results in locally small changes; however, the use of d2 as a stator and d1 as a flexor means that these small alterations are magnified at the isologous d1:d1 interface (Fig. 4A, C). This finding is supported by a broad survey of various proteins that found allostery to be correlated with the presence of isologous contacts[33,34]. The 3′ loop is situated opposite an interface helix (labelled i1 in Fig. 4), and together these elements slide past one another when the PTA domain transitions between the bound and empty forms. When superimposing the two states via d2, it is apparent that acetyl-CoA attracts the 3′ loop, pulling d1 and d2 closer together (Fig. 4B). Dyndom analysis[35] reveals this motion to be approximately a 10° rotation coupled with a 0.7 Å

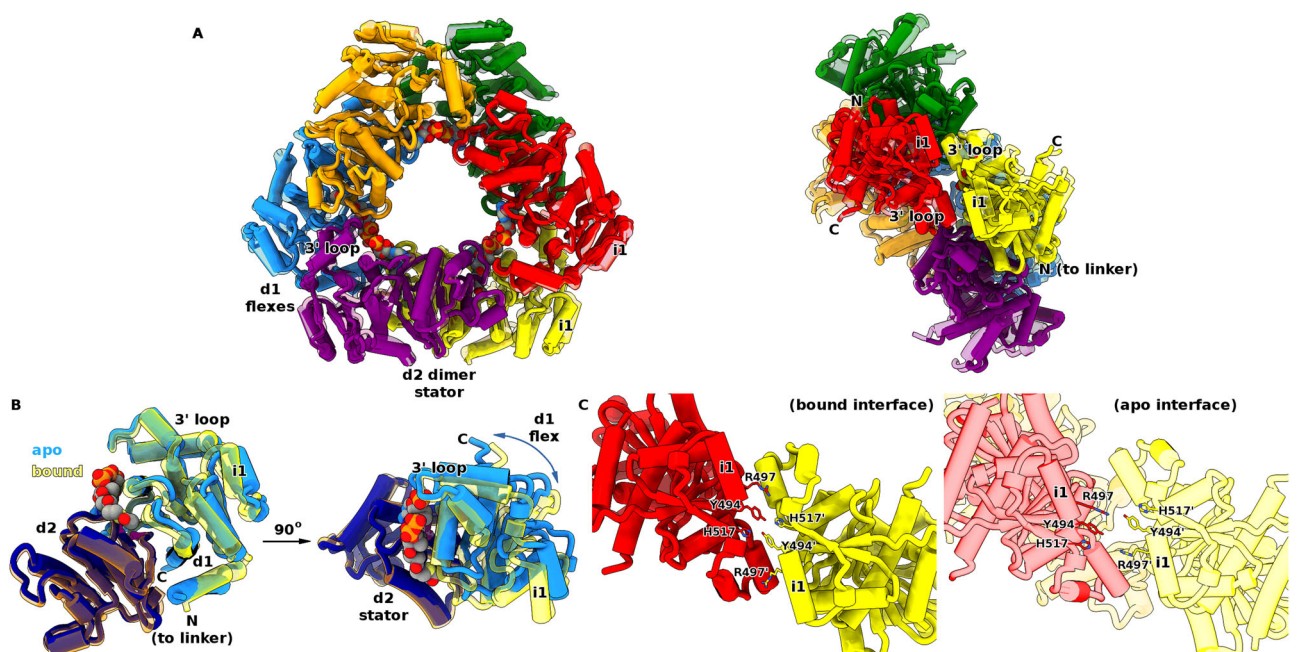

**Fig. 4 Conformational change between the bound and apo forms of the MaeB PTA domain. A** Overlay of the acetyl-CoA PTA$_{bound}$ (non-transparent) and PTA$_{apo}$ (transparent) structures, viewed from front and side. The d2:d2 dimer interface at the triangle middle edge acts as a stator where the d1 subdomain flexes relative to this, driven by shifts at the ligand-sensing 3′ loop at the d1:d1 interface (helix labelled i1 contributing Y494 and R497). **B** Demonstration of flexation in a single PTA monomer, from two orthogonal views. Acetyl-CoA "attracts" the 3′ loop, reducing the d1:d2 distance in the bound state (yellow d1/orange d2) relative to the apo form (light blue d1/dark blue d2). The d1 subdomain "rocker" movement (labelled d1 flex) transmits ligand occupancy at the 3′ loop to the interface (i1). Importantly the PTA N-terminal helix (linking to ME domain in full-length protein) shifts between the two states. **C** Zoomed-in view of isolated PTA$_{apo}$ and PTA$_{bound}$ d1:d1 interfaces, viewed from same orientation as that in (**A**) superimposition above. The interface helices shear relative to one another as the hexamer flexes, altering interactions between Y494, R497 and H517.

translation, in agreement with a 16° motion observed in active PTA enzymes[29]; the axis of flexation sits at the d1:d2 link at residues P595–A596 and L737–T738. The movement of d1 relative to d2 subtly remodels the hexamer and shifts the location of the N-terminal region that connects to the ME domain in full-length MaeB.

The importance of the 3′ loop in conveying the conformational change led us to examine a means by which we could prove the interface rearrangement was correlated to allostery. We hypothesised that 3′-dephosphoacetyl-CoA would not elicit inhibition as it would not attract the d1 3′ loop and remodel the hexamer; enzyme assays confirmed this (Fig. S5D), providing a non-mutagenic means to identify the signalling pathway between the PTA and ME domains. The role of hexamerization was further validated by an E544R obligate dimer variant, which was insensitive to inhibition by acetyl-CoA (Figs. 2D, S1C and Table S2).

**PTA:ME communication is long-range and mediated by a helical linker.** To couple our studies on the isolated domains to the mechanism utilised by the wt MaeB, it was imperative to obtain structural data on the full-length protein. We were successful in obtaining the acetyl-CoA bound inhibited state (MaeB$_{bound}$), diffracting to 2.72 Å resolution (Fig. 5A). Full-length MaeB$_{bound}$ retains the architectures observed for the isolated structures (Fig. 5B), placing the ME dimer twofold axis immediately below that of the PTA d2:d2 dimer axis. The folds are in excellent agreement with those observed in isolation (0.6 Å rmsd between ME domain structures, 1.2 Å between PTA domain structures). The large change in the hook subdomain (described in next section), results in smaller alterations in the swapped

active site helix and its local environment (see Discussion for a fuller interpretation of this). The resultant hexamer forms a disc measuring ~$100 \times 175 \times 195$ Å, with the ME domains twisted with respect to the triangular PTA edge. When measuring from the ME active site to the PTA 3′ loop, the two endpoints of the allosteric relay are spaced 60 Å apart in the shortest route (within a single monomer).

The individual ME and PTA domains are linked by two α-helices (Fig. 5C), the first, referred to here as L1, composed of a region unique to hybrid MEs (residues 422–437), and the second, L2, formed from the first α-helix of the PTA fold (residues 438–457). Together, these two regions can be seen to be a specialised adaptation to the hybrid protein sequences (Fig. S2). The N-terminus of L2 has a small kink centred around a motif formed of Q436-G437-P438. The relative orientation of the ME and PTA domains results in very few contacts between the two—L1 packs against the ME core via several conserved hydrophobic residues and L2 is ampipathic, with a hydrophobic N-terminus interacting with the PTA d1:d2 interface, and a polar C-terminus partly solvent-exposed (Fig. 5C). The ampipathic nature of L2 is conserved in amino acid type rather than absolute conservation of residues—I443, R444 and H451 displaying the highest degree of conservation, but no one position being invariable.

**Importance of the hybrid-specific hook subdomain in conferring allostery.** A key structural feature of the inhibited conformation of MaeB is the presence of a clash between the L2 helix and the ME domain surface, which results in the disordering of the ME hook subdomain (Fig. 5C). Disorder of the hook is apparent in all six chains of the hexamer, with small differences (chains A–D disordered from residues 371–383, residues 370–379

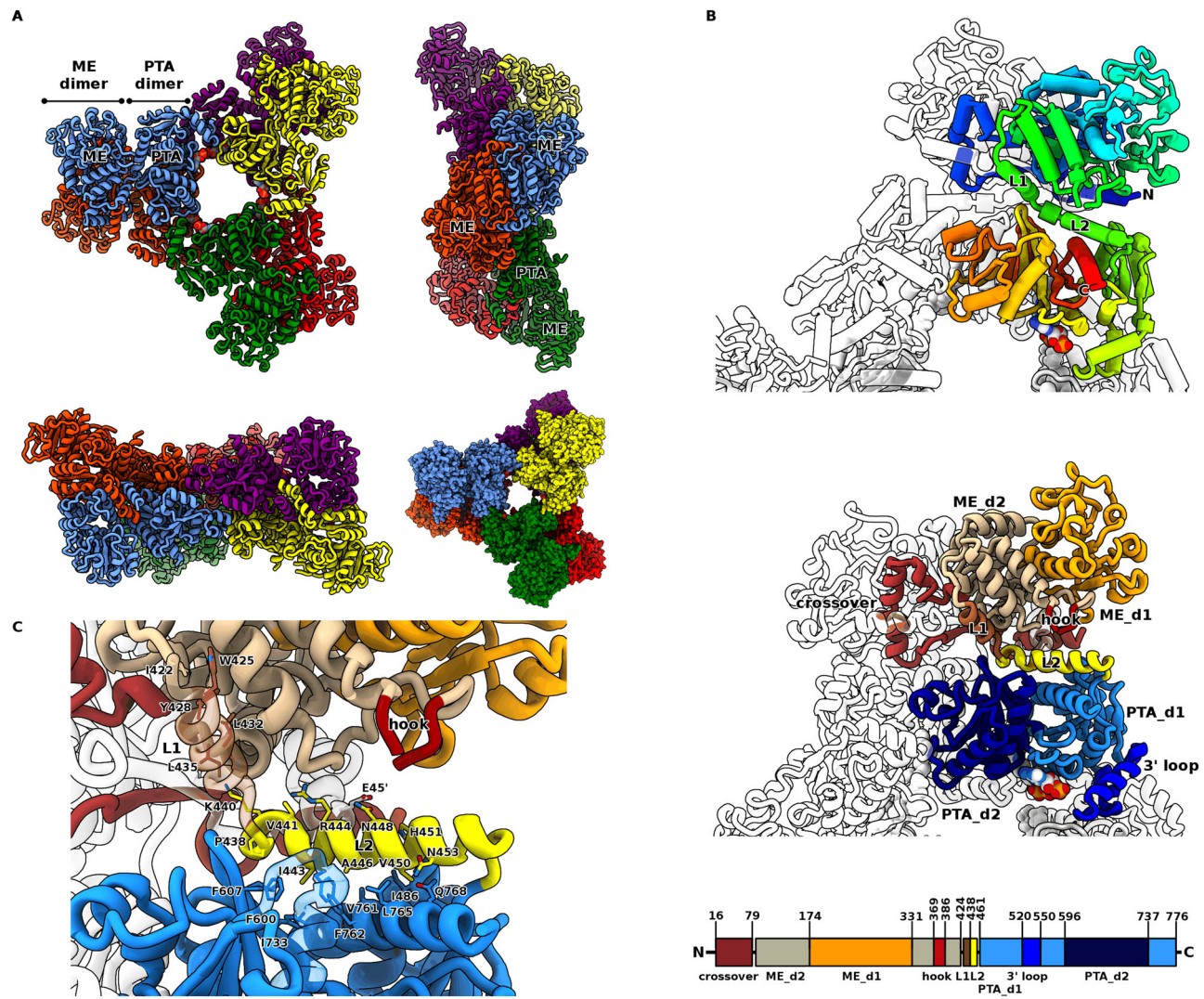

**Fig. 5 Structure of the full-length hybrid MaeB acetyl-CoA complex. A** The full-length ME:PTA MaeB protein retains the features of the isolated ME dimer and PTA hexamer, forming a central regulatory "hub" with the twofold axis of the enzymatic domains aligned with the PTA dimer axis. The hexamer is viewed from front, side and edge, with an additional front view in spacefill form demonstrating a clear groove/channel between the PTA and ME domains. The ME dimer is offset from the PTA "ring" edge by ~40°, as observed in the angle between the ME dimer and vertical axis in the upper-righthand panel. **B** Architecture of the hybrid MaeB enzyme: top panel, Jones rainbow (colouring from blue N-terminus to red C-terminus); middle panel, subdomains labelled and coloured identically to the figures of the isolated structures; lower panel, schematic of sequence elements. The ME and PTA domains are linked by two α-helices, labelled L1 (brown) and L2 (yellow), associated with the ME and PTA domain, respectively. **C** Residue interactions at the L1-L2 linker region. Part of L1 is rendered transparent for clarity, showing that L1 packs with several hydrophobic residues against the ME fold. Helix L2 makes hydrophobic interactions at the N-terminus, and more polar contacts at the C-terminal end. The placement of L2 disorders the ME hook subdomain and makes a small contact to the ME crossover region (at residue E45).

in chains E and F). Our observation of hook disorder in the inhibited state in tandem with hook conservation in hybrid enzymes led us to consider whether allostery was dependent on this feature. Using the minimal Phytoplasma enzyme as a guide[26], we made a chimeric MaeB that replaced the hook with the equivalent loop in non-hybrid enzymes (MaeB$_{\Delta hook}$, Fig. S7A). Kinetic assays confirmed that MaeB$_{\Delta hook}$ was insensitive to allosteric inhibition by acetyl-CoA (Fig. 2D), and excitingly, HPLC analyses demonstrated that acetyl-CoA was still bound at the PTA domain (Fig. S6B); hence removal of the hook sub-domain prevents signal transmission between the PTA and ME domains. Hook disorder and subsequent placement of L2 over the d1:d2 region observed to flex in the PTA$_{bound}$ to PTA$_{apo}$ transition provided a possible mechanism for direct PTA:ME communication and so we instigated crystallisation trials on Br⁻/urea washed full-length MaeB.

**The active state conformation reveals allostery is driven by large-scale enzyme rotation.** We used our Br⁻/urea protocol to successfully obtain a full-length apoprotein, MaeB$_{apo}$, from which we were able to determine a crystal structure at 3.70 Å resolution. Despite the limited resolution of this dataset, the electron density is readily interpretable, with good refinement statistics (Fig. S4 and Table S1). The structure was not solvable using the MaeB$_{bound}$ full-length model, necessitating the use of isolated MaeB$_{ME}$ and MaeB$_{PTA}$ search models, which results from large conformational differences between MaeB$_{bound}$ and MaeB$_{apo}$ (Fig. 6A–G and Supplementary Movie 1). MaeB$_{apo}$ preserves the hexameric D3 symmetry of MaeB$_{bound}$, but completely changes the placement of the ME dimer. Loss of acetyl-CoA causes a dramatic change to the ME:PTA domain relationship, mediated via a remodelling of the L1:L2 helix interaction (Fig. 6B). In MaeB$_{apo}$, L1 sits directly under L2 (helices in the same plane), replacing interactions made

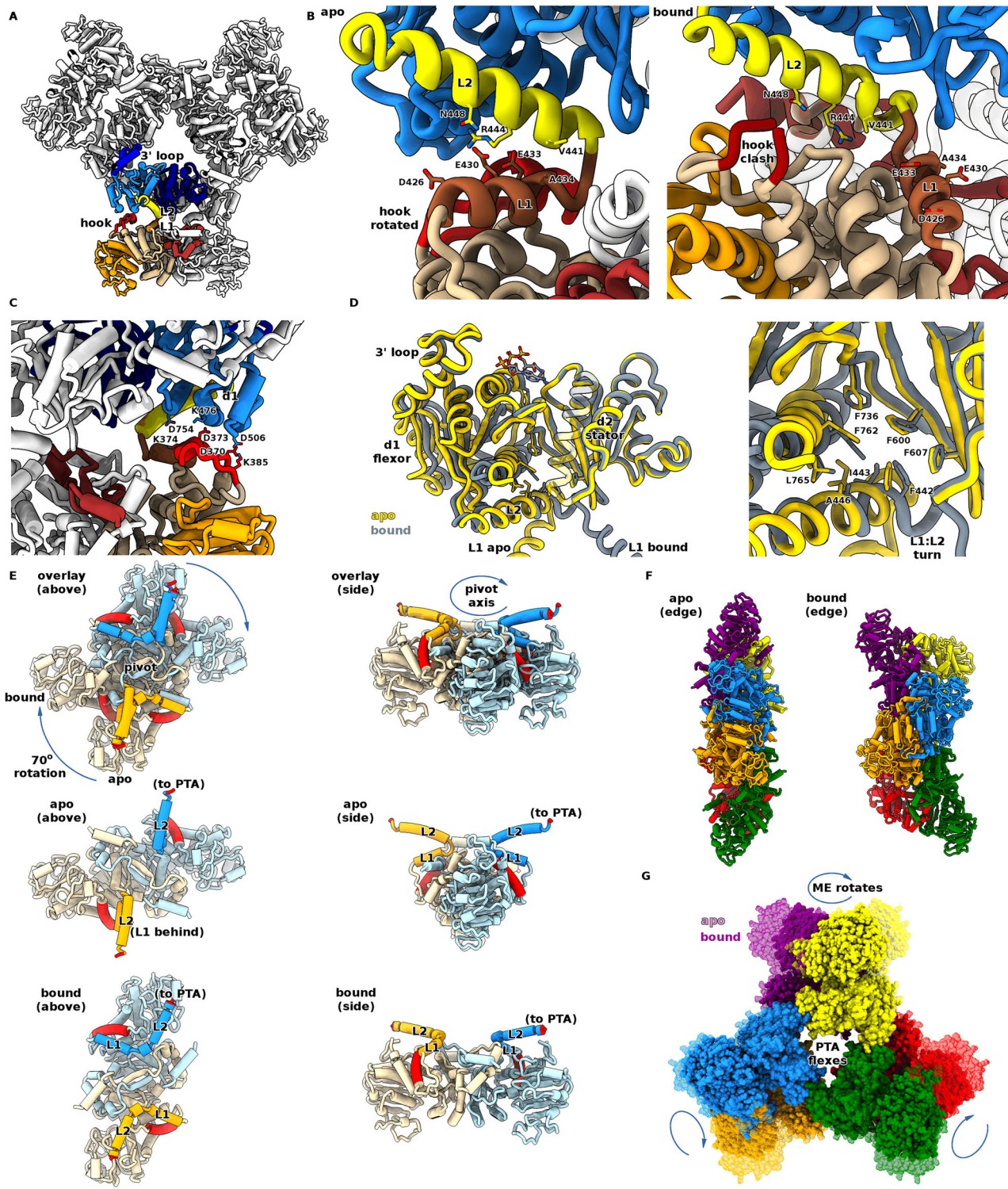

by the hook region in MaeB$_{bound}$ (helices ~120° apart). Thus, the MaeB$_{apo}$ structure reveals a requirement for L1:L2 complementarity, with hydrophobic contacts at the interhelical turn (A434:V441) and polar contacts along the helical faces (R444 and N448 to D426/E430/E433). MaeB$_{apo}$ has a greater degree of helicity at the L2 N-terminus, influenced by the conformations of F442/I443 which are part of a hydrophobic pocket between PTA d1 and d2 (Fig. 6D). The transitions between bound and apo structures of the PTA domain in the full-length protein agree with those observed previously for the isolated domain (a 10° d1:d2 flexation), but the consequences of altering the relative position of the PTA

C-terminal helix (L2) are magnified in the context of connection to the ME domain.

Dyndom analysis reveals that the apo to bound transition is represented by a 69° relative rotation of the ME dimer, with a 3.4 Å translation that pulls the ME dimer away from the PTA regulatory hub (Fig. 6E). Resulting from this large-scale rotation, the ME hook subdomain is freed from the clash with L2 observed in MaeB$_{bound}$, and is now located under PTA d1 where it is able to adopt a helical conformation identical to that determined for the isolated ME domain structure. The upper face of the hook makes several polar contacts to the lower face of d1, forming

**Fig. 6 Structure of full-length apo-MaeB reveals a large-scale rotation governs allostery. A** Structure of MaeB$_{apo}$. The PTA hexameric "hub" retains the conformation observed in the isolated PTA apo form, and the ME dimer rotates relative to this to place the hook subdomain under the PTA d1 region. This rotation is enabled by a different L1:L2 juxtaposition that places L1 directly below L2. **B** Comparison of the apo (left) and bound (right) linker regions. The helical character of this region is retained in MaeB$_{apo}$, but an alteration in the loop between L1 and L2 results in the lower face of L2 (V441, R444, N448) interacting with L1 (acidic patch of D426, E430, E433) rather than the interactions with the ME domain observed in MaeB$_{bound}$. **C** The rotation of the ME dimer in MaeB$_{apo}$ allows the hook subdomain to adopt a folded conformation, wherein it interacts with PTA d1. **D** Comparison of PTA conformational change between MaeB$_{apo}$ (yellow) and MaeB$_{bound}$ (grey). The d1:d2 flexation is essentially similar to that observed for the isolated PTA structures, but with the additional outcome that the linker region is remodelled in response to changes at the d1:d2 hydrophobic core (zoomed in righthand panel). **E** Mechanism of rotation of ME domain, viewed from two separate orientations. A direct comparison of MaeB$_{apo}$ and MaeB$_{bound}$, with an overlay obtained from PTA superposition (removed for clarity), reveals that the PTA:linker (labelled "to PTA", coloured red) is in the same relative location in both forms but the ME domain pivots 70° about the twofold axis. The L-shaped L1:L2 orientation in MaeB$_{bound}$ is replaced by the co-linearity of MaeB$_{apo}$, swinging the ME fold relative to the pivot axis—a representative ME feature below the linker (α-helix 400–414) is coloured red to assist in the interpretation of this movement. Also refer to Supplementary Movie 1. **F** Resultant difference in shape of the MaeB$_{apo}$ and MaeB$_{bound}$ hexamers. Rotation of the ME dimers in the apo form results in a more streamlined disc-shape, given that the ME domains now align with the PTA hexamer d2:d2 edge. **G** Overlay of MaeB$_{apo}$ (transparent) and MaeB$_{bound}$. Flexation at the PTA hub rotates three ME dimers at each flat edge of the triangular array.

the sole PTA:ME contacts not mediated by the linker helices (Fig. 6C). The enzyme activity of the isolated ME domain is less than the full-length enzyme (Table S2), indicating that the MaeB$_{apo}$ PTA:ME interactions assist in promoting catalysis. When L1 moves relative to L2, the ME dimer rotates on its twofold axis under the PTA twofold axis (Fig. 6E). Remarkably, the L2:PTA link remains fixed (compare red regions in Fig. 6E), resulting in the ME dimer effectively turning like a propeller on the PTA trimer flat edge (Fig. 6E–G). This 69° rotation approaches the maximal shift of 90° possible in a dimeric configuration (91° clockwise being identical to 89° counter clockwise). The net effect of this rotation is to "pull" the ME dimer into alignment with the PTA dimer, resulting in a thinner/ streamlined hexamer, 74 Å wide versus 90 for the bound state (Fig. 6F). ME:PTA contacts are more pronounced in the bound structure, hence the 3.4 Å translation along the twofold axis may help pull the ME dimer clear of the PTA domain as it rotates; the lack of extensive ME:PTA hydrogen bonding is presumably a consequence of avoiding an energetic barrier to the bound:apo conformational change.

## Discussion

The structure of MaeB and its constituent domains provides clear information as to the adaptations required to meld the ME enzyme and PTA sensory domain into a long-range allosteric complex. In MaeB, the PTA domains assemble into a unique hexameric form (no other hexameric PTA domain assemblies were found using TopSearch[36]). Assembly into a D3 hexamer provides a large, co-operative interface for the PTA subunits to communicate occupancy to one another; indeed, dihedral symmetry has been found to be enriched in key metabolic enzymes wherein isologous interfaces impose conformational change on opposing subunits[33,34]. The large number of contacts across all three PTA interfaces (d1:d1, d1:d2, d2:d2) adapt the PTA dimer into a highly co-operative hexamer. A conformational change across all subunits (rather than individual monomers switching independently) is in keeping with the Monod–Wyman–Changeaux model of allostery[37], with transition between two endpoint states that preserve the inherent symmetry—an inactive T and active R state. Our bound and apo structures thus represent these T and R states that were first proposed in the studies by Sanwal et al. on *E. coli* MaeB[38]. The most significant adaptations to the ME enzymatic domain in the hybrid enzymes are the inclusion of the hook subdomain and an extra helix at the C-terminus that forms the L2 connection to the PTA domain.

The clear placement of acetyl-CoA at the PTA domain informs on the likely mode of action of active PTA enzymes that inter-convert acetyl-CoA and phosphate with CoA and acetyl phosphate. The position of acetyl-CoA in MaeB is different to that of characterised PTA enzyme complexes[30,31] but is in excellent agreement with biochemical data[30,39,40]. Hence, MaeB$_{bound}$ represents a closed form of the domain that can be used to model likely interactions in catalytically competent homologues.

Our structures provide a clear description as to how acetyl-CoA occupancy is communicated 60 Å to the ME domain. Acetyl-CoA binding attracts the 3′ loop of the PTA domain, in turn bringing d1 and d2 relatively closer together. The relatively small 10° flexation drives hydrophobic rearrangement at the interface, which in turn influences the hydrophobic N-terminal end of the L1 linker helix (residues F442, I443 and A446). The PTA core:linker interaction thus affords a mechanism for altering the environment of the L1:L2 linker, such that L1:L2 juxtaposition is the dominant difference between the MaeB$_{apo}$ and MaeB$_{bound}$ forms. Two features dictate the resulting conformational change —firstly, the L2 C-terminus:PTA relationship remains largely identical between the bound and apo forms (Fig. 6B) and secondly, there are few direct contacts between the ME and PTA domains that limit their positional freedom. Hence, the L1:L2 helices behave akin to the coupling rods of a steam train wheel, resulting in free rotation of the ME domain (Fig. 6E). The magnitude of this rotation, being ~70°, is remarkable given the modest perturbations seen between the T and R states in model allosteric enzymes like pyruvate kinase (16° rotation)[41], fructose-1,6-bisphosphatase (15º rotation)[42] and aspartate carbamoyl-transferase (12° rotation)[43]. Our observations on MaeB$_{apo}$ are based on a single, static crystal form; this likely samples from a wider distribution in solution.

Eukaryotic MEs, despite a completely different ATP-sensitive allosteric domain, use a largely identical active site with MaeB subdomains d1 and d2 analogous to domains C and B. Human ME adopts at least three states: an open form competent to bind substrate and release product, a closed inhibited state and a tightly closed quaternary complex that shields the active site from solvent and orders the catalytic residues in a productive con-formation[44]. The similarity between our observed binary ME: NADP$^+$ complex and that of the eukaryotic enzymes (Fig. S3A) infers a similar requirement for a closed state. Using the distance between d2 D150 and d1 N300 as a measure of closure, our isolated ME enzyme (12.2 Å) and full-length structure (13.2 Å) are closer to the human enzyme open state (PDB: 1QR6, D256: N421 = 12.3 Å) as opposed to the closed state (PDB: 1EFK, D256: N421 = 9.2 Å). Our kinetic assays demonstrate that the place-ment of the unique hook subdomain on MaeB d1 is key to gating activity. Interestingly, residue F386 at the base of the hook is an equivalent position to human ME residue L507, which has been

shown to be involved in the conversion to the closed catalytically active state. It is likely that our inhibited MaeB$_{bound}$ structure represents an inactive state by both disordering the local hook structure and limiting the dynamics of d1. Reordering of the hook domain, and rotation of the ME dimer, as observed in the MaeB$_{apo}$ structure, provide a conformation where d1 has relatively more freedom (also aided by the 3.4 Å translation of the ME dimer away from the PTA central hub). Goodsell and Olson describe two possible mechanisms for allosteric regulation of dihedral rings—relative rotation or a pincher-like motion[33]; our structures of MaeB suggest a third amalgamated possibility wherein flexation (pincher-like movement) of the PTA domain results in co-ordinated rotation of the ME component. This would then licence an additional motion of ME closure. The domain-swapped nature of the MaeB ME dimer is also involved in the regulatory mechanism. The swapped α-helix from the opposing monomer contributes the catalytic tautomerization residue Y52 to the active site and is bordered by the conserved polar residue E45 (Fig. S8). This feature has very different environments in the two states, located next to the semi-conserved L2 linker residue H451 in MaeB$_{bound}$, and next to the hook subdomain residue Y379 in MaeB$_{apo}$ (Fig. S8). Hence, the linker and hook, the two features that differ most in the two states, have the potential to influence active site geometry in different ways. In total, the regulatory mechanism involves communicating ligand occupancy at the PTA sensor to the ME active site. A major rotation and minor translation of the enzyme relative to the central hub relieves interactions that repress the ability of the ME subdomains to open and close, which in turn licences enzyme mobility and thus catalysis.

The precise role of MaeB in *Bdellovibrio bacteriovorus* is not currently documented. However, it is of interest that *Bdellovibrio* has no homologue of the proton-translocating transhydrogenase and so the concerted action of MaeB, pyruvate carboxylase and malate dehydrogenase may act to transfer reducing equivalents between NAD(H) and NADP(H)[1]. *Bdellovibrio* MaeB may be of particular importance when nutrients flow from prey to predator, *e.g.* amino acids entering the TCA cycle and eventually producing NADPH and pyruvate; these can then feed gluconeogenesis and fatty acid biosynthesis, acting to store energy between kills. In this regard the allosteric inhibitor acetyl-CoA would act in the role of feedback inhibition. Several MaeB homologues have different susceptibilities to allosteric effectors, which likely arise from PTA domain sequence variation. *S. meliloti* TME is not inhibited by either CoA or acetyl-CoA[24]; our model reveals that this could be elicited by the presence of a repulsion-conferring glutamic acid residue at MaeB R581, in proximity to the nucleotide pyrophosphate, and lack of basic 3′ phosphate interacting residues (MaeB R535 and K538). *S. meliloti* DME can be inhibited by acetyl-, propionyl- and butyryl CoA, providing another example of the versatility of the PTA domain in the hybrid enzyme allostery[23]. The allosteric plasticity of the appended PTA domain is also apparent in MaeB from the nitrogen-fixing bacterium *Azospirillum brasilense*, which is apparently regulated by the acetyl-CoA: CoA ratio[31]; this enzyme differs from the other characterised MaeB enzymes via a Y576F substitution (*B. bacteriovorus* numbering), located at the acetyl pocket of the PTA domain. The means by which the allosteric PTA domain communicates to the ME enzyme via use of linker helix rearrangement and contact to the hook region is likely to be common to all MaeB enzymes, but the precise accommodation of allosteric effector at the binding pocket may be tuneable through amino acid variation in differing homologues.

Our work illustrates how the PTA enzyme fold has been repurposed and utilised as a sensory domain to control the activity of hybrid MEs. Our analysis of variant enzyme forms (point mutants and MaeB$_{\Delta hook}$ construct) may have utility in biosynthetic applications that require high MaeB activity, given their ability to circumvent inhibition by acetyl-CoA[45–47]. Moreover, we demonstrate how the unique structural features of hybrid MEs (the Hook and L1 & L2 helices) function mechanistically during allosteric regulation. This work provides a basis to understand regulation in this group of enzymes, which are positioned at a crucial metabolic junction.

## Methods

**Cloning and mutagenesis.** The original gene annotation for MaeB in *Bdellovibrio bacteriovorus* strain HD100 had been misannotated as two genes (*bd1833* and *bd1834*) on NCBI, due to a sequencing read error producing a premature stop codon. Thus throughout this manuscript the start codon of *bd1834* and the stop codon of *bd1833* were used as the coding extent of MaeB. Full-length MaeB (Met1–Lys780), ME domain only (MaeB$_{ME}$, Met1–Ala434) and PTA domain only (MaeB$_{PTA}$, Ser439–Lys780) constructs were amplified from HD100 genomic DNA (kind gift from Sockett laboratory, University of Nottingham). The amplified genes were inserted in-frame into a pET28a plasmid, immediately after the N-terminal His$_6$ tag sequence, using restriction free cloning. Mutant proteins (R535A, R535E, N718D, E544R) and chimeric MaeB$_{\Delta hook}$ were produced by standard site directed mutagenesis; primer sequencing for cloning and mutagenesis are provided in Table S3. Constructs and mutations were confirmed by sequencing, before transformation into the *E. coli* expression strain BL21λDE3.

**Protein production and purification.** Cells containing MaeB constructs were grown at 37 °C (shaken at 180 rpm) in Terrific Broth supplemented with 100 μg/ml kanamycin until an OD$_{600}$ of 0.8 was reached. Gene expression was induced with 1 mM IPTG and incubated O/N at 18 °C (shaken at 180 rpm). Cells were harvested by centrifugation at 6891 RCF for 10 min. The supernatant was discarded and the pellets were frozen at −20 °C for storage. Cells were re-suspended into fresh lysis buffer (50 mM HEPES pH 7.5, 20 mM Imidazole pH 7.5, 300 mM NaCl, 0.05% w/v Tween-20) and incubated with ~1 mg/ml lysozyme for 30 min at 4 °C. Cells were lysed using sonication. Insoluble cell debris was removed by centrifugation for 1 h at 32,826 RCF. The soluble material was passed over a HisTrap HP column (GE Healthcare life science), pre-equilibrated with lysis buffer. The column was washed with 20 column volumes (CV) of lysis buffer to remove non-specific interacting proteins. The column was then eluted with 10 CV of elution buffer (50 mM HEPES pH 7.5, 300 mM NaCl, 400 mM Imidazole pH 7.5). Pooled fractions were transferred to 10 kDa MWCO dialysis tubing and dialysed against 2 L of dialysis buffer (20 mM BisTris pH 6.0, 75 mM KCl, 5 mM MgCl$_2$, 2 mM DTT for full-length MaeB and MaeB$_{ME}$ constructs and 20 mM HEPES pH 7.5, 300 mM NaCl for PTA-only constructs), for 3 h at 4 °C. The dialysis tubing was then transferred to a fresh 2 L of dialysis buffer, O/N at 4 °C to fully equilibrate. Proteins were finally concentrated to >10 mg/ml.

**Size-exclusion chromatography.** Size-exclusion chromatography was carried out using a 26/600 superdex 200 prep grade column (GE healthcare life science) attached to an AKTA purifier system operated using Unicorn software. A 26/600 superdex 200 column with a volume of 320 ml was used for high resolution of proteins with molecular weights between 10 kDa and 600 kDa. The column was pre-equilibrated with 1.5 CV of protein dialysis buffer.

**Structure determination.** Crystals were grown at 18 °C using the sitting drop vapour diffusion technique. Table S4 lists the crystallisation conditions used for each construct. Crystals were transferred to cryoprotection solution (Mother liquor + 20% v/v ethylene glycol) and incubated for 5 min before flash cooling in liquid nitrogen. MaeB$_{bound}$ full-length crystals were harvested 2 days after the first crystals developed; reduced growth time resulted in better quality diffraction. MaeB$_{ME}$: NADP$^+$ complex crystals were obtained by soaking into a solution of cryoprotection liquor supplemented with 1 mM NADP$^+$ and incubated for 1 h before being flash cooled in liquid nitrogen.

Diffraction data were collected at Diamond light source in Oxford, UK. The data were autoprocessed by ISPyB automated pipelines (Xia2 and autoproc). PTA-only was initially solved by molecular replacement using PHASER and a pruned 1TD9 search model[48,49]. The structure of MaeB$_{ME}$ was solved using a 5CEE (*Candidatus* Phytoplasma AYWB-ME) dimer search model[50,51]. Full-length MaeB (bound and apo forms) were solved using our individual refined ME and PTA domains as search models. Manual model building and refinement were performed using COOT, Phenix and PDB-redo[52].

**Enzyme assays.** Spectrophotometric assays were used to monitor the oxidative decarboxylation of L-malate to pyruvate and the reductive carboxylation of pyruvate to L-malate using a CARY 300 Bio, UV-Visible spectrophotometer operated by CARY WinUV kinetics software. All assays recorded the Δabsorbance at 340$_{nm}$ (NADPH) at 25 °C. A molar extinction coefficient for NADPH of 6.22 mM$^{-1}$ cm$^{-1}$

# ARTICLE

was used for all calculations. Assays were initiated by the addition of enzyme to the otherwise complete reaction mixture.

Oxidative decarboxylation reactions were performed using a standard reaction mixture: 100 mM Tris-HCl pH 7.5, 50 mM KCl, 5 mM $MnCl_2$, 5 mM L-malate, 0.5 mM $NADP^+$. Apparent Michaelis–Menten parameters were determined for both substrate (L-malate) and cofactor ($NADP^+$), by varying the concentration of one substrate (or cofactor) about its $K_M$, whilst other components were left unchanged at saturating concentrations. A maximum concentration of 10 mM L-malate was used in assays, beyond which reduced/inhibited rates were observed. A final enzyme concentration of 100 nM was used to determine $K_M$(malate), whilst 50 nM was used to determine $K_M$($NADP^+$). Assays using MaeB$_{ME}$ and MaeB$_{\Delta hook}$ variants required 1 µM of enzyme to compensate for lower activity. The curves were fitted to Michaelis–Menten model $v_0 = V_{max} * $[malate]$/(K_M + $[malate]$)$, where $v_0$ is the initial rate. $k_{cat}$ was obtained by adjusting the model to compensate for enzyme concentration (Et), $v_0 = Et * k_{cat} *$[malate]$/(K_M + $[malate]$)$. All kinetic parameters were fit with non-linear regression using GraphPad Prism 7 software and presented as means of triplicate experiments ± standard error. Substrate inhibition data was fit to the equation $v_0 = (V_{max} * S)/(K_M + S*(1 + (S^h)/K_i))$ using Sigmaplot.

The reductive carboxylation reactions were measured by the reduction in absorbance at 340 nm in 100 mM Tris-HCl pH 7.5, 50 mM KCl, 5 mM $MnCl_2$, 30 mM $NaHCO_3$, 0.1 mM NADPH and 10 mM pyruvate. The assay was initiated by the addition of enzyme (final enzyme concentration of 1 µM).

**Inhibitor studies**. A range of known ME effectors were tested against MaeB activity under standard reaction conditions: 100 mM Tris-HCl pH 7.5, 50 mM KCl, 5 mM $MnCl_2$, 0.5 mM L-malate and 0.5 mM $NADP^+$. The compounds (acetyl phosphate, pyruvate, succinate, fumarate, oxaloacetate, glucose-6-phosphate, glutamate and glutamine) were tested at 2 mM, whilst CoA and acetyl-CoA were tested at 5 µM. The results are presented as a percentage of the measured activity in the presence of effector compared to the measured activity of MaeB under standard reaction conditions.

The mode of CoA and acetyl-CoA inhibition was investigated by measuring substrate velocity curves (L-malate and $NADP^+$) at multiple inhibitor concentrations. Curves were fitted to the non-competitive inhibition model $V_0 = (V_{max}/(1 + [I]/K_i)) * $[malate]$/(K_M + $[malate]$)$. Data were also plotted as a double reciprocal plot (Lineweaver–Burk) for illustrative purposes. CoA or acetyl-CoA inhibitor potency was investigated by measuring the activity of ligand free MaeB$_{apo}$ under standard assay conditions at multiple inhibitor concentrations. The data were graphically plotted as rate versus [inhibitor] and fitted using non-linear regression to the [Inhibitor] versus response—Variable slope model on GraphPad Prism 7 software.

**Reverse phase HPLC**. A modified version of a previously described HPLC method for CoA and acetyl-CoA detection was used to identify co-purifying ligands[53]. Protein samples were prepared by boiling 100 µM protein samples for 10 min and then centrifuging at high speed (20,000 x $g$) to remove insoluble material (denatured protein). The supernatant was transferred into HPLC micro-sampling vials (Thermo Scientific). HPLC analysis was performed using a Ultimate3000 UHPLC by ThemoFisher Scientific, equipped with a Kinetex 150 × 2.1 mm C18 column; detection procedure: 5 µl injection, 0.175 ml/min flow rate, 20 min run time, column oven 25 °C, UV detection 259 nm. HPLC grade acetonitrile was added at a ratio of 6 (acetonitrile) to 94 (mobile phase) to prepare the mobile phase (100 mM monosodium phosphate and 75 mM sodium acetate adjusted to pH 4.6 with phosphoric acid) before use. CoA and acetyl-CoA gave distinct and well-separated peaks using this method, eluting at ~3.5 ml and ~7 ml respectively (Fig. S6D). Baseline shifts were observed with new mobile phase formulations, thus a standard was chromatographed alongside each new batch for comparison. The system was calibrated with CoA and acetyl-CoA standards ranging between 10–150 µM (chromatographed together and separately to determine the identity of each peak).

**Acetyl-CoA removal (washing protocol)**. The co-purifying CoA and acetyl-CoA ligands were removed by the addition of a mild denaturing wash step to the purification procedure. Protein was bound to a 1 ml HisTrap column and washed with 20 CV of lysis buffer to remove contaminants. The column was then further washed with a mild denaturing buffer (2.5 M Urea, 2.5 M KBr, 50 mM HEPES pH 7.5) for 100 CV at a flow rate of 1 ml/min. Another 20 CV of lysis buffer was passed over the column to remove the denaturing agents before the column was eluted with 10 CV of elution buffer. Eluted fractions were then pooled and dialysed as described for other samples.

**Reporting summary**. Further information on research design is available in the Nature Research Reporting Summary linked to this article.

## Data availability

Coordinates and structure factors have been deposited in the PDB under accession codes 6ZNJ (full-length apo form), PDB: 6ZNG (full-length acetyl-CoA bound form), PDB: 6ZN4 (MaeB$_{ME}$ apo form), PDB: 6ZN7 (MaeB$_{ME}$ $NADP^+$ complex), PDB: 6ZN9

(MaeB$_{PTA}$ apo form), PDB: 6ZNT (MaeB$_{PTA}$ acetyl-CoA bound form), PDB: 6ZNR (MaeB$_{PTA}$ R535A), PDB: 6ZNE (MaeB$_{PTA}$ R535E), PDB: 6ZNK (MaeB$_{PTA}$ N718D) and PDB: 6ZNU (MaeB$_{PTA}$ E544R). Source data are provided with this paper for Fig. 2, Sup Fig. 1, Sup Fig. 5, Sup Fig. 6, Table S2 in the Source Data file. Source data are provided with this paper.

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

## Acknowledgements

We thank members of the University of Birmingham structural biology community for discussions, especially those within the Lovering group. We thank the Sockett group University of Nottingham for discussions on *Bdellovibrio* biology. We gratefully acknowledge Diamond synchrotron source for access to x-ray beamtime. We thank Clarissa Czekster at the University of St. Andrews for assistance with enzyme kinetic data analysis. This project was funded by BBSRC studentship 1500753 to C.J.H. and a BBSRC David Phillips fellowship to P.J.M. (BB/S010122/1).

## Author contributions

Conceptualisation, C.J.H. and A.L.L.; Investigation, C.J.H., I.T.C., P.J.M. and A.L.L.; Formal analysis, C.J.H. and A.L.L.; Writing, C.J.H. and A.L.L.; Funding acquisition, C.J.H. and A.L.L.; Resources, C.J.H., I.T.C., P.J.M. and A.L.L.; Supervision, I.T.C., P.J.M. and A.L.L.

## Competing interests

The authors declare no competing interests.
