## [Peer Review File · Nature Communications]

REVIEWER COMMENTS

Reviewer #1 (Remarks to the Author):

Lovering and coauthors report structural and biochemical studies of a hybrid malic enzyme from *B. bacteriovorus*, Bd-MaeB. They have determined the structures of the dimeric ME domain and the hexameric PTA domain, revealing how the PTA domain recognizes the potent allosteric inhibitor acetyl-CoA. They have also determined the structures of full-length MaeB hexamer, inhibited by acetyl-CoA (2.7 Å resolution) and in the active form, without acetyl-CoA (3.7 Å). Mutagenesis and kinetic studies have been carried out to assess the structural observations. Overall, the reported results represent major accomplishments, indicating how conformational changes in the PTA domain upon acetyl-CoA binding is transmitted to the ME domain, and enhancing our understanding of allosteric regulation of MaeB, and enzymes in general.

Many of the figure panels are too small to visualize the details in them. In addition, in many panels the structure of the protein dominates, due to the use of this 'worm' representation, even though the panels are meant to show other aspects of the structure, for example ligand binding.

A large rotation is observed for the ME dimer relative to the PTA hexamer upon acetyl-CoA binding. It is not described in the text whether there are conformational changes in the active site region of ME in the acetyl-CoA bound structure compared to the free enzyme, except an overall rmsd of 0.6 Å (line 246). A figure showing an overlay of the active site region would be important. How does the rotation of ME dimer relative to PTA regulates catalysis if there are no major changes in the active site region? If there are changes, how are they triggered by this rotation? A better analysis/description of the molecular mechanism of this allosteric inhibition would strengthen the paper.

CoA has 45-fold weaker activity compared to acetyl-CoA (Fig. 2E,F). The recognition of the acetyl group is not clearly described in the text or figure (Fig. 3B). In addition, the only residue mentioned as in contact with the acetyl group, Q683 (line 177), is not conserved in other MaeB homologs (Fig. S2).

An N718D mutant was tested (Fig. 2C), but it is not clear where this residue is located in the binding site (Fig. 3B). In addition, this residue is Arg in *E. coli* MaeB (Fig. S2). The authors suggest this mutant has higher activity because it does not have copurified acetyl-CoA. Why do the R535 mutants have lower activity? They do not have copurified acetyl-CoA either (Fig. S6C).

Along the lines the two questions above, the sequence conservation for the PTA domain is much weaker compared to the ME domain (Fig. S2). Many of the side chains shown in Fig. 3B for binding acetyl-CoA are not conserved in other MaeB homologs (Fig. S2). In addition, the PTA domain is responsible for mediating activation by Glu, Asp and other compounds for the *E. coli* MaeB. Is the Bd-MaeB also activated by these compounds?

The kinetic data clearly show substrate inhibition by malate (Fig. S5C). What is the K_i of this inhibition? Is it comparable to the potency of oxaloacetate? Could the two compounds use the same binding site in the ME domain to inhibit the enzyme?

I believe the ME_d1 and d2 labels in the bar graph at the bottom of Fig. 5B are swapped. It is not clear why d2 precedes d1 in the primary sequence of ME, while d1 precedes d2 in PTA.

An inactive R state and an active T state (line 334): I believe the authors have gotten the R and T states swapped.

Reviewer #2 (Remarks to the Author):

In this manuscript, the authors show the mechanistic basis for regulation of hybrid PTA-MEs. The structural and kinetic analyses reported here demonstrate the allosteric regulation of the ME:PTA coupling, and the drastic conformational changes of this protein during allosteric binding. In my opinion, this work represents a huge amount of work involved in generating the data to test the allosteric coupling taking place in this interesting type of MEs. It is rare to find such a comprehensive piece of work involving site directed mutagenesis, construction of truncated proteins and x-ray crystallography of several different recombinant proteins, combined with kinetic data. Overall, I have no major concerns with the findings and interpretation of the data reported in this manuscript. I feel that the work is of good quality and that the conclusions are basically robust. Below, I indicate some concerns and considerations for the authors.

- Line 15 & 17: Hybrid enzyme and PTA meaning should be explained in the abstract.
- Line 35: It is not clear why the authors indicate that NAD-MEs function as anaplerotic. Several NAD-MEs are unable to catalyse the reductive carboxylation of pyruvate; instead, they mostly function as generating pyruvate as fuel for the Krebs cycle, instead of forming intermediates. The reductive carboxylation of pyruvate was measured in some NADP-ME isoforms.
- Line 44: plastidic
- Line 53: are predicted
- Line 109: Why the authors indicate that this difference is small? In my opinion, it represents a great difference to other MEs which crystal structure were resolved (and was only shown only by Phytoplasma ME until present).
- Line 122: a series (a)
- Lines 125-126: The authors were able to measure the pyruvate reductive carboxylation reaction in MaeB. What happened in the case of the MaeBapo? Although this activity is extremely low, it may be relevant for malate synthesis in some metabolic situation. Was this activity inhibited by acetylCoA?
- Table S2: There is a mistake; Km values for NADP are indicated as mM, they should be in microM.
- Table S2: Which kinetic parameters were considered statistically different? The authors should indicate this type of statistical analyses in this Table.
- Line 128 & Figure S5C: The authors show inhibition of ME activity by malate concentrations higher than 10 mM. This inhibition is also observed in some NADP-MEs from C4 plants in a pH-dependent way.
- Lines 140-150: The authors suggest CoA and acetyl-CoA act as non-competitive inhibitors. However, from Lineweaver-Burk plots (Figure S5) this is not obvious; instead, it seems they behave as mixed-type inhibitor in plots where malate was fixed. Moreover, MaeB (with acetyl-CoA) and MaeBapo Km for NADP are different, as shown in Table S2.
- The authors were able to generate an apoprotein with the Br-/urea wash. However, is it possible that this treatment could lead to some conformational change besides the elimination of acetyl CoA?
- The authors conclude that PTA-MaeB switches between two endpoint states. However, could the Br-/urea wash be so drastic and avoid the identification of some conformational changes of individual monomers switching independently?
- Lines 692 & 706: There are two mistakes, please correct: reductive carboxylation & oxidative decarboxylation.

Reviewer #3 (Remarks to the Author):

This paper focuses on the ME enzymes, a family of central metabolic enzymes that play key roles at the nexus of glycolysis and the citric acid cycle. Specifically, it focuses on MaeB, a larger, prokaryotic, "hybrid" ME with unknown structure. Acetyl-CoA and other important metabolites were known to allosterically regulate MaeB in a way that requires its additional PTA domain. However, the structure of MaeB and the regulatory mechanism involving acetyl-CoA were unknown. This paper combines structural biology, mutagenesis, enzyme kinetics assays, and other biophysical experiments to answer these questions. The authors report high-resolution crystal structures of the isolated domains, and

moderate-to-low-resolution crystal structures of full-length protein. The structures reveal an unexpected hexameric complex enabled by the PTA domain, as well as the notably distal acetyl-CoA binding site. Further comparisons of apo vs. acetyl-CoA-bound structures highlight a seemingly novel structural mechanism for allosteric regulation involving large-scale quaternary rearrangements. Along the way, mutations to key binding-site residues, exploration of several metabolites and carefully chosen mimic molecules, enzyme kinetics assays, and other biophysical experiments are used judiciously to establish connections between structural models and functional interpretations.

MAJOR COMMENTS:

Overall this is a very nice paper. It focuses on an important system that sits at the nexus of glycolysis and the citric acid cycle, so is metabolically central. The structures reported are generally of sufficiently high quality to justify the conclusions (see one caveat below). Previously, structures of hexameric ME enzymes were not known, so this paper reports novel findings in that regard. Stemming from these structures is an interesting mechanism of allostery involving large-scale rotations within the hexamer. While not novel in a general sense, given that other oligomeric systems undergo coordinated rotations in response to allosteric effectors (as the paper discusses), knowledge of such allostery in this particular, previously unknown protein architecture is new. The text is reasonably well written, and the figures are generally quite pretty and informative. Overall, we think that this will be a paper worthy of publication once a few relatively minor issues are addressed (see below).

Although the paper overall is quite good, we do have a few moderate criticisms. First, in lines 360-363, the authors emphasize the magnitude of the hexamer rotations (~ 70 degrees) as being distinct from what is seen in other oligomeric allosteric systems. However, the precise angular changes from apo- to effector-bound may differ significantly in solution vs. in the crystal lattice. Moreover, the rotations seen in solution and in the cell likely sample from a distribution that is wider than the very tight distribution seen from these structural data and inferred from the static crystal structure models. This possibility should be emphasized, and the language pertaining to the magnitude of these rotations should be toned down.

A second moderate criticism is that we found it difficult to fully assess the quality of the experimental structures, since the validation reports provided to reviewers are incomplete for several entries. In particular, the full-length MaeB WT ("native") apo structure at 3.7 Å resolution has some stats in Table S1 that are not as good as for all the other structures (which look quite good). For example, the merging statistics are almost an order of magnitude worse than other structures, including the other low-resolution (>3 Å) structure, PTA N718D. Also, the Rfree-Rwork gap of $\sim 7\%$ is a bit large relative to the gap for all other structures. Unfortunately, we are unable to further investigate the quality of the data and model for this dataset, since the validation report states "The reported resolution of this entry is unknown" and thus is blank for many parts of the report.

Overall, this is a solid paper with contributions that others in the field will find useful and interesting. We believe that it merits publication, after correction of said issues.

MINOR COMMENTS:

Line 65: acetyl-CoA generator

But Fig. 1A does not show ME as directly making acetyl-CoA.

Do you mean that ME is needed indirectly to make acetyl-CoA, by making malate into pyruvate, which is then used by another enzyme (PD) to make acetyl-CoA?

Line 136: missing "AND CoA" (right?)

Line 142: Global fitting to different inhibition models suggested CoA and acetyl-CoA act as non-competitive inhibitors (V_{max} decreased and the apparent K_M remained unaffected), which is

supported graphically by Lineweaver-Burk plots (Figure S5)

But in the Lineweaver-Burk plots, "graphically" V_{max} looks the same for CoA and basically the same for acetyl-CoA, and so does K_m for that matter. That said, it is clear V_{max} is changing in the E,G,I panels (Michaelis-Menten plots). Did you extract V_{max} and K_m values from the data? Are those reported somewhere?

Line 145: However, K_i values derived from Michaelis-Menten inhibition plots, where NADP⁺ was fixed, have a large degree of error, due to the restriction of available L-malate concentrations, leading to ambiguity of V_{max} values at high inhibitor concentrations.

Does this help explain the above (about interpreting Lineweaver-Burk plots)? Given that V_{max} is unreliable, are the conclusions above about the non-competitive mechanism (which rely on a certain V_{max} effect) still reliable? Maybe they are, if the V_{max} precise numbers are uncertain but the trends are still obvious...

Line 163: the hexameric assembly unique to MaeB

That's not necessarily true -- could be that previous structures of other PTA protein(s) just had differences in the construct or lattice form, but they can actually form hexamers in cells/solution.

Line 196: Variants R535E and R535A abolished acetyl-CoA binding, and generated an enzyme insensitive to allosteric inhibition by both CoA and acetyl-CoA (Figure 2C and Table S2), although these mutants retained similar activity to wild-type (wt) MaeB

This is more like 70%, so not really "similar". Rerword.

Line 206: HPLC analyses confirmed we were able to generate an apoprotein via this strategy (Figure S6C)

Do you mean Fig. S6B? That is where apo is introduced...

Line 236: The role of hexamerization was further validated by an E544R obligate dimer variant, which was insensitive to inhibition by acetyl-CoA (Figures 2D, S1C and Table S2).

I'm confused by this. We get that E544R makes a dimer (not hexamer), and is insensitive to acetyl-CoA. But Table S2 shows that E544R has a much lower k_{cat}/K_m than WT -- yet Fig. 2D suggests it has the same % activity as WT. Is there a difference in construct we are just missing? Or is this an issue related to normalization of % activity for each mutant in Fig. 2? Specifically, is the normalization different in Fig. 2C vs. Fig. 2D? It seems that in Fig. 2C the data for each construct may be normalized to WT, whereas for Fig. 2D the data for each construct may be normalized to the "no effector" state for that mutant (not to WT)... If so, that is weird -- and needs to be clearly explained in the caption! In general, we found the % activity definition(s) in the paper to be lacking, and thus confusing.

Line 101: Comparatively, the hook region is easily discerned as an insert in the primary sequence of hybrid MEs (Figure S2)

and

Line 254: Together, these two regions can be seen to be a specialized adaptation to the hybrid protein sequences (Figure S2).

We don't see how that is discernible from this sequence alignment figure, since only hybrid ME sequences are shown, and not any other MEs or reference sequences for comparison... What are we missing?

Line 265: A key structural feature of the inhibited conformation of MaeB is the presence of a clash between the L2 helix and the ME domain surface, which results in the disordering of the ME hook subdomain (Figure 5C). Disorder of the hook is apparent in all six chains of the hexamer, with small differences (chains A-D disordered from residues 371-383, residues 370-379 in chains E and F). Could be good to show this in a supp figure: strong density before/after, but no density for hook. Not required, though.

Line 329: The large number of contacts across all three PTA interfaces (d1:d1, d1:d2, d2:d2) adapt the PTA dimer into a highly co-operative hexamer. A conformational change across all subunits (rather than individual monomers switching independently) is in keeping with the Monod-Wyman-Changeaux model of allostery (Monod J., 1965), with transition between two endpoint states that preserve the inherent symmetry – an inactive R and active T state.

Does this imply positive cooperativity for acetyl-CoA binding (by analogy to positive cooperativity for O₂ binding with Hb in the MWC model)? We did not notice any experiments in this paper that explicitly address the existence of cooperative binding in this system. At least discuss.

Line 334: inactive R and active T state

Traditionally for Hb, R = “active” meaning it is O₂-bound, and T = “inactive” meaning it is unbound... Is this definition/nomenclature different for other allosteric proteins that are enzymes?

Line 349: Our multistate structures

Reword... “multistate structures” to many people will mean something more like models from qFit or Phenix ensemble refinement.

Please clarify why MaeB_PTA in the apo vs. acetyl-CoA-bound forms crystallize in such different conditions. Was it not possible to use the same conditions for both to minimize differences between the datasets? (This would be perfectly reasonable BTW.)

Fig. 1 caption: “redbrown” - not sure what that means...

Fig. 1 structures are a bit too busy in our opinion -- should be clipped more tightly (true of some other figures too, like Fig 3D-E).

Fig. 1 could show rotation arrows to indicate which view angle is which

Fig. 2: add titles to panels so don't have to look back at caption all the time

Fig. 2 caption (& S5): how much of each effector (uM)? “Saturating amounts”? Or at least “see Methods”.

A/B/C etc. panel labels are quite small (too small?) in some figures.

Fig. 5B bottom (sequence diagram) could be moved up to Fig. 1A, before structures. As-is, we had to look forward to Fig. 5B several times before it was even introduced.

Fig S1C: void = what? Unfolded due to mutation?

Fig. S6 should say what is being monitored on the y-axis. We had to dig into the Methods to see that this wavelength (apparently) monitors acetyl-CoA. In general, could include a bit more in the captions about what we're seeing -- or at least redirect to Methods in key spots.

Figure S7: Should show where the green vs. blue parts are in the structure overlay!

Table S1: X-ray stats table: Need to say what #s in () mean.

Table S1 does not list the PDB IDs, despite the facts that they are listed elsewhere in the paper and that Table S1 has an empty row for them...

Table S3 (crystallization conditions) seems to have an error in merging the first two rows, leading to initial confusion about which condition goes with which structure.

Line 53 typo

Line 122 typo

Finally, PLEASE PUT FIGURES AND CAPTIONS IN-LINE NEAR THE RELEVANT PORTIONS OF THE TEXT!
Or at the very least, PUT THE FIGURE CAPTIONS NEXT TO THE FIGURES! It makes the reviewers' jobs much harder when we have to juggle 3-4 different PDFs and Word docs on-screen at once... If this formatting is due to journal policy rather than author choice, PLEASE CHANGE THE POLICY!

We review non-anonymously,
Daniel Keedy
Virgil Woods

REVIEWER COMMENTS

Reviewer #1 (Remarks to the Author):

Lovering and coauthors report structural and biochemical studies of a hybrid malic enzyme from *B. bacteriovorus*, Bd-MaeB. They have determined the structures of the dimeric ME domain and the hexameric PTA domain, revealing how the PTA domain recognizes the potent allosteric inhibitor acetyl-CoA. They have also determined the structures of full-length MaeB hexamer, inhibited by acetyl-CoA (2.7 Å resolution) and in the active form, without acetyl-CoA (3.7 Å). Mutagenesis and kinetic studies have been carried out to assess the structural observations. Overall, the reported results represent major accomplishments, indicating how conformational changes in the PTA domain upon acetyl-CoA binding is transmitted to the ME domain, and enhancing our understanding of allosteric regulation of MaeB, and enzymes in general.

We thank the reviewer for noting our work/results as “major accomplishments”, and the relation to understanding diverse enzymes in general.

Many of the figure panels are too small to visualize the details in them. In addition, in many panels the structure of the protein dominates, due to the use of this ‘worm’ representation, even though the panels are meant to show other aspects of the structure, for example ligand binding.

We appreciate the need to show more detail on ligand binding, and have introduced a new supplementary figure for acetyl-CoA binding, inserted as new FigS7 (with subsequent renumbering of existing S7 to become S8). This includes details on both faces of acetate group recognition.

We have tried to keep the protein “centreplace” with a deep field of view because we feel that it allows tracking of which fold elements contribute to the conformational change mechanism when comparing figures / subsections. We are happy to split (and thus enlarge) any of the figures and so increase figure number if appropriate (via editorial direction).

A large rotation is observed for the ME dimer relative to the PTA hexamer upon acetyl-CoA binding. It is not described in the text whether there are conformational changes in the active site region of ME in the acetyl-CoA bound structure compared to the free enzyme, except an overall rmsd of 0.6 Å (line 246).

We agree that some changes are described somewhat minimally in the results (due to the reason that they are structurally similar), but have a whole subsection dedicated to the important changes in the hook region of the enzyme (“Importance of the hybrid-specific hook subdomain in conferring allostery”) and have now added a statement that directs readers to the discussion where other small changes in the active site provide a possible explanation for regulation of catalysis. This now reads “The folds are in excellent agreement with those observed in isolation (0.6 Å rmsd between ME domain structures, 1.2 Å between PTA domain structures). The large change in the hook subdomain (described in next section), results in smaller alterations in the swapped active site helix and its local environment (see Discussion for a fuller interpretation of this).”

A figure showing an overlay of the active site region would be important. How does the rotation of ME dimer relative to PTA regulates catalysis if there are no major changes in the active site region?

In lines 366-394 of the Discussion, we use the strong structural homology to the eukaryotic enzymes to propose a similar mechanism of regulation (catalysis regulated by domain opening/closure), but mediated by a different means (the appended PTA domain).

If there are changes, how are they triggered by this rotation? A better analysis/description of the molecular mechanism of this allosteric inhibition would strengthen the paper.

We have added a stronger statement to “cap and summarize” the above explanation, and this section now ends “In total, the regulatory mechanism involves communicating ligand occupancy at the PTA sensor to the ME active site. A major rotation and minor translation of the enzyme relative to the central hub, relieves interactions that repress the ability of the ME subdomains to open and close, which in turn licenses enzyme mobility and thus catalysis.”

CoA has 45-fold weaker activity compared to acetyl-CoA (Fig. 2E,F). The recognition of the acetyl group is not clearly described in the text or figure (Fig. 3B).

In addition, the only residue mentioned as in contact with the acetyl group, Q683 (line 177), is not conserved in other MaeB homologs (Fig. S2).

We have added a statement to better explain the recognition, noting (i) partial conservation, and (ii) that Q683 is one of several residues that contact the acetate moiety. This now reads “A pocket formed from Y576, T614, N718, Y721, K722, P735 and Q748 surrounds the acetate moiety; a single acetyl-binding residue, Q683, is provided by the opposing monomer of the dimer. These residues exhibit some variation in homologues (Figure S2)”.

An N718D mutant was tested (Fig. 2C), but it is not clear where this residue is located in the binding site (Fig. 3B).

This context is now provided in the additional figure S7 (noted above), detailing the AcCoA acetate-binding pocket.

In addition, this residue is Arg in *E. coli* MaeB (Fig. S2). The authors suggest this mutant has higher activity because it does not have copurified acetyl-CoA. Why do the R535 mutants have lower activity? They do not have copurified acetyl-CoA either (Fig. S6C).

The different mutant activities are dependent on structural context, which has effects beyond a binary copurify/non-copurify with allosteric ligand phenotype. The R535 mutants remove the arginine hydrophobic contribution to a pocket of L531/L534/I540/L548, whereas the N718D mutant is isosteric; these will differentially effect protein stability and dynamics, hence activity.

Along the lines the two questions above, the sequence conservation for the PTA domain is much weaker compared to the ME domain (Fig. S2). Many of the side chains shown in Fig. 3B for binding acetyl-CoA are not conserved in other MaeB homologs (Fig. S2). In addition, the PTA domain is responsible for mediating activation by Glu, Asp and other compounds for the *E. coli* MaeB. Is the Bd-MaeB also activated by these compounds?

We investigated a number of known *E. coli* effectors against the activity of Bd-MaeB (see Supplementary Figure 5D) and did not see an identical profile to Ec-MaeB. For instance, Glutamate and Glucose-6P which activated Ec-MaeB full-length and not Ec-MaeB ME domain, had no effect upon Bd-MaeB. We hope that our work on Bd-MaeB serves as a framework for further studies on diverse enzymes.

The kinetic data clearly show substrate inhibition by malate (Fig. S5C). What is the K_i of this

inhibition? Is it comparable to the potency of oxaloacetate? Could the two compounds use the same binding site in the ME domain to inhibit the enzyme?

The K_i for substrate inhibition was calculated using a modified substrate inhibition equation (see updated methods section on line 717) and the K_i value of 39.46 ± 19.2 mM has been added to Supplementary Figure 5C. The focus of our investigation centred on understanding the regulatory mechanism invoked by acetyl-CoA interaction with the distal PTA domain. As a result, we did not further investigate inhibition by oxaloacetate as this is likely to occur by the same (known) mechanism as in other malic enzymes [1, 2].

I believe the ME_d1 and d2 labels in the bar graph at the bottom of Fig. 5B are swapped. It is not clear why d2 precedes d1 in the primary sequence of ME, while d1 precedes d2 in PTA.

We apologise for this switch in the ME_d1 and ME_d2 labels, and have now corrected these in a new version of Figure 5. We also have added a statement to line 101 of the ME fold description that explains the unusual ordering thus: "This nomenclature identifies d1 as the main functional domain of the enzyme, but means that d2 precedes it in the amino acid sequence".

An inactive R state and an active T state (line 334): I believe the authors have gotten the R and T states swapped.

We apologise for this error and have amended the text to keep R active, T inactive as standard.

Reviewer #2 (Remarks to the Author):

In this manuscript, the authors show the mechanistic basis for regulation of hybrid PTA-MEs. The structural and kinetic analyses reported here demonstrate the allosteric regulation of the ME:PTA coupling, and the drastic conformational changes of this protein during allosteric binding. In my opinion, this work represents a huge amount of work involved in generating the data to test the allosteric coupling taking place in this interesting type of MEs. It is rare to find such a comprehensive piece of work involving site directed mutagenesis, construction of truncated proteins and x-ray crystallography of several different recombinant proteins, combined with kinetic data.

We are very pleased that the reviewers acknowledge this effort and thank them for their appreciation of it.

Overall, I have no major concerns with the findings and interpretation of the data reported in this manuscript. I feel that the work is of good quality and that the conclusions are basically robust. Below, I indicate some concerns and considerations for the authors.

- Line 15 & 17: Hybrid enzyme and PTA meaning should be explained in the abstract.

We take note of this recommendation and have reworded the abstract to "MaeB grouping, multidomain" to explain the hybrid nature, and used the longhand form of PTA "phosphotransacetylase".

- Line 35: It is not clear why the authors indicate that NAD-MEs function as anaplerotic. Several NAD-MEs are unable to catalyse the reductive carboxylation of pyruvate; instead, they mostly function as generating pyruvate as fuel for the Krebs cycle, instead of forming intermediates. The reductive carboxylation of pyruvate was measured in some NADP-ME isoforms.

We agree that the reviewer uses a better summarizing definition for the NAD-dependent enzymes. As such, we have amended the text to read "In general, NAD⁺ dependent MEs function to

provide pyruvate for the TCA cycle”.

- Line 44: plastidic

We have made this change.

- Line 53: are predicted

We have made this change.

- Line 109: Why the authors indicate that this difference is small? In my opinion, it represents a great difference to other MEs which crystal structure were resolved (and was only shown only by Phytoplasma ME until present).

We apologise for the understatement and have removed the word “small”.

- Line 122: a series (a)

We have now altered this to “a series of”.

- Lines 125-126: The authors were able to measure the pyruvate reductive carboxylation reaction in MaeB. What happened in the case of the MaeBapo? Although this activity is extremely low, it may be relevant for malate synthesis in some metabolic situation. Was this activity inhibited by acetyl CoA?

The k_{cat}/K_M values for the reductive carboxylation reaction were 0.3% of that for the oxidative decarboxylation reaction, suggesting the equilibrium lies far towards producing pyruvate. We therefore focussed our investigation on the favourable oxidative decarboxylation reaction, likely to dominate in-vivo [3].

- Table S2: There is a mistake; Km values for NADP are indicated as mM, they should be in microM.

This has been amended in our new revised Table S2.

- Table S2: Which kinetic parameters were considered statistically different? The authors should indicate this type of statistical analyses in this Table.

A statistical test was not performed to deduce differences between the kinetic parameters of different enzyme forms. We deduced a parameter was different if it was larger than the expected experimental error.

- Line 128 & Figure S5C: The authors show inhibition of ME activity by malate concentrations higher than 10 mM. This inhibition is also observed in some NADP-MEs from C4 plants in a pH-dependent way.

Inhibition by malate is known for several MEs; we are happy to include a reference for a C4 study if the reviewers think it will direct the reader. As noted above, we have now included the K_i of L-malate at line 717.

- Lines 140-150: The authors suggest CoA and acetyl-CoA act as non-competitive inhibitors.

However, from Lineweaver-Burk plots (Figure S5) this is not obvious; instead, it seems they behave

as mixed-type inhibitor in plots where malate was fixed. Moreover, MaeB (with acetyl-CoA) and MaeB_{apo} K_M for NADP are different, as shown in Table S2.

The restriction of L-malate concentrations used in our assay means a degree of error was observed as a consequence of this limitation. As a result, we are careful not to use these data to obtain K_i values for CoA or acetyl-CoA and instead compare potency through IC₅₀ experiments. This section (lines 144 – 153) has been reworded to emphasise this point. The K_i value was only reported for the fixed [NADP⁺] experiment where a non-competitive model is appropriate and graphical plots (Figure S5.I) clearly show a decrease in V_{max} and an unchanged K_M.

Global fitting suggests a non-competitive inhibition model and mixed as similarly appropriate when compared, for fixed [malate] (where the data is more robust). The alpha value for mixed inhibition model (0.590) suggests the model is more similar to a non-competitive model (lying closer to 1). We directly know that an uncompetitive model is not appropriate from our crystal structure in complex with acetyl-CoA, without bound substrates. We also presume the mechanism of inhibition holds true for both substrates, hence a non-competitive inhibition model, which is appropriate when [malate] is fixed should hold true for when [NADP⁺] is fixed.

We consider the differences in NADP K_M between MaeB and MaeB_{apo} to be within reasonable experimental error.

- The authors were able to generate an apoprotein with the Br-/urea wash. However, is it possible that this treatment could lead to some conformational change besides the elimination of acetyl CoA?

The similar structures derived from the Br-/urea procedure and three separate mutants (line 215, “demonstrate the same features”), coupled with increased enzymatic activity in the full-length enzyme, lead us to believe that no artefactual outcome occurred.

- The authors conclude that PTA-MaeB switches between two endpoint states. However, could the Br-/urea wash be so drastic and avoid the identification of some conformational changes of individual monomers switching independently?

In all the mutant forms (R535E, R535A, N718D) and the Br-/urea wash, all 24 monomers retain a similar pose that argues against independent switching; the three extensive interfaces (figure 3A) provide strong co-operativity between oligomers.

- Lines 692 & 706: There are two mistakes, please correct: reductive carboxylation & oxidative decarboxylation.

These mistakes have been corrected on the manuscript.

Reviewer #3 (Remarks to the Author):

This paper focuses on the ME enzymes, a family of central metabolic enzymes that play key roles at the nexus of glycolysis and the citric acid cycle. Specifically, it focuses on MaeB, a larger, prokaryotic, “hybrid” ME with unknown structure. Acetyl-CoA and other important metabolites were known to allosterically regulate MaeB in a way that requires its additional PTA domain. However, the structure of MaeB and the regulatory mechanism involving acetyl-CoA were unknown. This paper combines structural biology, mutagenesis, enzyme kinetics assays, and other biophysical experiments to

answer these questions. The authors report high-resolution crystal structures of the isolated domains, and moderate-to-low-resolution crystal structures of full-length protein. The structures reveal an unexpected hexameric complex enabled by the PTA domain, as well as the notably distal acetyl-CoA binding site. Further comparisons of apo vs. acetyl-CoA-bound structures highlight a seemingly novel structural mechanism for allosteric regulation involving large-scale quaternary rearrangements. Along the way, mutations to key binding-site residues, exploration of several metabolites and carefully chosen mimic molecules, enzyme kinetics assays, and other biophysical experiments are used judiciously to establish connections between structural models and functional interpretations.

MAJOR COMMENTS:

Overall this is a very nice paper. It focuses on an important system that sits at the nexus of glycolysis and the citric acid cycle, so is metabolically central. The structures reported are generally of sufficiently high quality to justify the conclusions (see one caveat below). Previously, structures of hexameric ME enzymes were not known, so this paper reports novel findings in that regard. Stemming from these structures is an interesting mechanism of allostery involving large-scale rotations within the hexamer. While not novel in a general sense, given that other oligomeric systems undergo coordinated rotations in response to allosteric effectors (as the paper discusses), knowledge of such allostery in this particular, previously unknown protein architecture is new. The text is reasonably well written, and the figures are generally quite pretty and informative. Overall, we think that this will be a paper worthy of publication once a few relatively minor issues are addressed (see below).

We thank the reviewers for their comments, particularly regarding establishing a connection between the structures and functional interpretation.

Although the paper overall is quite good, we do have a few moderate criticisms. First, in lines 360-363, the authors emphasize the magnitude of the hexamer rotations (~70 degrees) as being distinct from what is seen in other oligomeric allosteric systems. However, the precise angular changes from apo- to effector-bound may differ significantly in solution vs. in the crystal lattice. Moreover, the rotations seen in solution and in the cell likely sample from a distribution that is wider than the very tight distribution seen from these structural data and inferred from the static crystal structure models. This possibility should be emphasized, and the language pertaining to the magnitude of these rotations should be toned down.

We have added a statement that reminds the reader that our observations are “in-crystallo” (to the end of this section): “Our observations on MaeB_{apo} are based on a single, static crystal form; this likely samples from a wider distribution in solution.”

A second moderate criticism is that we found it difficult to fully assess the quality of the experimental structures, since the validation reports provided to reviewers are incomplete for several entries. In particular, the full-length MaeB WT (“native”) apo structure at 3.7 Å resolution has some stats in Table S1 that are not as good as for all the other structures (which look quite good). For example, the merging statistics are almost an order of magnitude worse than other structures, including the other low-resolution (>3 Å) structure, PTA N718D. Also, the R_{free}-R_{work} gap of ~7% is a bit large relative to the gap for all other structures. Unfortunately, we are unable to further investigate the quality of the data and model for this dataset, since the validation report states “The reported resolution of this entry is unknown” and thus is blank for many parts of the report.

We were actually pleasantly surprised by the quality of the 3.7 Å map (indeed, see Figure S4), and note that the Rfactor gap is not representative of its interpretability (we only had the opportunity to shoot one sample of this form thus far, no doubt several samples would potentially yield an improvement on this). Nevertheless, we are able to model into this data unambiguously, and draw conclusions that are independent of its relative low resolution (the conformational change is dependent on secondary structure alteration and domain movement rather than amino acid rotamers). We would be happy to share this data with the reviewers if necessary. The apo-to-bound transition of the low-resolution full-length protein is substantiated in part by identical movements in the higher resolution PTA-only structures; which also validates our approach of investigating both individual domains and the full-length protein in tandem.

Overall, this is a solid paper with contributions that others in the field will find useful and interesting. We believe that it merits publication, after correction of said issues.

MINOR COMMENTS:

Line 65: acetyl-CoA generator

But Fig. 1A does not show ME as directly making acetyl-CoA.

Do you mean that ME is needed indirectly to make acetyl-CoA, by making malate into pyruvate, which is then used by another enzyme (PD) to make acetyl-CoA?

We apologise for any confusion here and have added “, the latter made indirectly” to this description on line 65.

Line 136: missing “AND CoA” (right?)

Yes, the reviewers are correct – it is also insensitive to inhibition by CoA; the text has been altered to make this finding clear. “inhibition by acetyl-CoA, CoA and acetyl phosphate”.

Line 142: Global fitting to different inhibition models suggested CoA and acetyl-CoA act as non-competitive inhibitors (V_{max} decreased and the apparent K_M remained unaffected), which is supported graphically by Lineweaver-Burk plots (Figure S5)

But in the Lineweaver-Burk plots, “graphically” V_{max} looks the same for CoA and basically the same for acetyl-CoA, and so does K_M for that matter. That said, it is clear V_{max} is changing in the E,G,I panels (Michaelis-Menten plots). Did you extract V_{max} and K_M values from the data? Are those reported somewhere?

Please see the comment to reviewer 2 that also addresses this concern.

Simply fitting the inhibition data at fixed $[NADP^+]$ to the Michaelis-Menten equation is ambiguous for higher inhibitor concentrations and does not help analysis. So extracted V_{max} and K_M values make little sense when $[NADP^+]$ is fixed. Yet V_{max} clearly decreases and K_M values stay similar when [malate] is fixed.

Line 145: However, K_i values derived from Michaelis-Menten inhibition plots, where $NADP^+$ was fixed, have a large degree of error, due to the restriction of available L-malate concentrations, leading to ambiguity of V_{max} values at high inhibitor concentrations.

Does this help explain the above (about interpreting Lineweaver-Burk plots)? Given that V_{max} is unreliable, are the conclusions above about the non-competitive mechanism (which rely on a certain V_{max} effect) still reliable? Maybe they are, if the V_{max} precise numbers are uncertain but the trends

are still obvious...

As noted in our comments to the other reviewers, we presume the mechanism of inhibition holds true for both substrates, hence a non-competitive inhibition model, which is appropriate when [malate] is fixed (figure S5.I) should hold true for when [NADP⁺] is fixed. We do not extract a K_i value from the fixed [NADP⁺] data due to ambiguity of the fit and instead opted to compare potency of inhibitors through the IC₅₀ experiments.

Line 163: the hexameric assembly unique to MaeB

That's not necessarily true -- could be that previous structures of other PTA protein(s) just had differences in the construct or lattice form, but they can actually form hexamers in cells/solution.

We firstly defined residues that assist in MaeB hexamer interface formation and then looked for these in PTA sequences using the PDBeMotif (<https://www.ebi.ac.uk/pdbe-site/pdbemotif/>) server, finding no hits. We then used the sequence-independent TopSearch server (<https://topsearch.services.came.sbg.ac.at/>) to check across all PDB ASU and biological arrays and didn't find any hexamer. Hence, at present, we can be confident that no verified hexameric PTA proteins have been identified – this may of course differ in due time.

Line 196: Variants R535E and R535A abolished acetyl-CoA binding, and generated an enzyme insensitive to allosteric inhibition by both CoA and acetyl-CoA (Figure 2C and Table S2), although these mutants retained similar activity to wild-type (wt) MaeB

This is more like 70%, so not really "similar". Reword.

We agree and have now reworded this to read "these mutants retained ~70% activity to".

Line 206: HPLC analyses confirmed we were able to generate an apoprotein via this strategy (Figure S6C)

Do you mean Fig. S6B? That is where apo is introduced...

This has been corrected in the text.

Line 236: The role of hexamerization was further validated by an E544R obligate dimer variant, which was insensitive to inhibition by acetyl-CoA (Figures 2D, S1C and Table S2).

I'm confused by this. We get that E544R makes a dimer (not hexamer), and is insensitive to acetyl-CoA. But Table S2 shows that E544R has a much lower kcat/Km than WT -- yet Fig. 2D suggests it has the same % activity as WT. Is there a difference in construct we are just missing? Or is this an issue related to normalization of % activity for each mutant in Fig. 2? Specifically, is the normalization different in Fig. 2C vs. Fig. 2D? It seems that in Fig. 2C the data for each construct may be normalized to WT, whereas for Fig. 2D the data for each construct may be normalized to the "no effector" state for that mutant (not to WT)... If so, that is weird -- and needs to be clearly explained in the caption! In general, we found the % activity definition(s) in the paper to be lacking, and thus confusing.

Initially we normalised the data for MaeB, E544R and Δ hook within each enzyme type without inhibitor compared to when inhibitor was added, to make any differences between +/- inhibitor more apparent. We have now normalised the data to the same singular control for all datasets (normalised to MaeB under normal reaction conditions), for consistency. We have also added "% of activity datasets are normalised to the activity of MaeB under normal reaction conditions" to the legend of Figure 2.

Line 101: Comparatively, the hook region is easily discerned as an insert in the primary sequence of hybrid MEs (Figure S2)

and

Line 254: Together, these two regions can be seen to be a specialized adaptation to the hybrid protein sequences (Figure S2).

We don't see how that is discernible from this sequence alignment figure, since only hybrid ME sequences are shown, and not any other MEs or reference sequences for comparison... What are we missing?

This is true, and we apologise for the omission. To make this more clear, we have constructed a new Figure S2 that includes other ME sequences, and thus makes the adaptation easy to identify.

Line 265: A key structural feature of the inhibited conformation of MaeB is the presence of a clash between the L2 helix and the ME domain surface, which results in the disordering of the ME hook subdomain (Figure 5C). Disorder of the hook is apparent in all six chains of the hexamer, with small differences (chains A-D disordered from residues 371-383, residues 370-379 in chains E and F). Could be good to show this in a supp figure: strong density before/after, but no density for hook. Not required, though.

It is difficult to “show” a feature that loses density/definition; we will leave this to user inspection of the deposited maps. We were truly unable to locate/model the hook subdomain in the inhibited forms, even having tried SA-omit, feature-enhanced and sharpened maps in Phenix.

Line 329: The large number of contacts across all three PTA interfaces (d1:d1, d1:d2, d2:d2) adapt the PTA dimer into a highly co-operative hexamer. A conformational change across all subunits (rather than individual monomers switching independently) is in keeping with the Monod-Wyman-Changeaux model of allostery (Monod J., 1965), with transition between two endpoint states that preserve the inherent symmetry – an inactive R and active T state.

Does this imply positive cooperativity for acetyl-CoA binding (by analogy to positive cooperativity for O₂ binding with Hb in the MWC model)? We did not notice any experiments in this paper that explicitly address the existence of cooperative binding in this system. At least discuss.

We use our structural data to imply likely positive co-operativity between the units of the hexamer, with each protomer making contacts to 4 other chains. We did attempt ITC experiments to investigate binding cooperativity. However, we were unable to fit the data unambiguously - possibly due to the inherent complexity of the hexamer and/or the presence of residual Coa/acetyl-coa bound to the protein which affected the ‘known’ starting state. Hence, without a method to reliably obtain 100% apoprotein (our crystallization of this form may select for this without needing a completely pure sample state), we believe the structural data is more apt/illustrative of this concept (lines 332-336).

Line 334: inactive R and active T state

Traditionally for Hb, R = “active” meaning it is O₂-bound, and T = “inactive” meaning it is unbound... Is this definition/nomenclature different for other allosteric proteins that are enzymes?

We apologise and as outlined in response to reviewer 1, have now standardized this in the text.

Line 349: Our multistate structures

Reword... “multistate structures” to many people will mean something more like models from qFit or

Phenix ensemble refinement.

We appreciate the potential ambiguity with the qFit approach and other multimodel ensemble approaches. We have therefore reworded this to “Our structures”.

Please clarify why MaeB_PTA in the apo vs. acetyl-CoA-bound forms crystallize in such different conditions. Was it not possible to use the same conditions for both to minimize differences between the datasets? (This would be perfectly reasonable BTW.)

Yes, this is impossible given incompatibility between the two (conformational change, resulting different unit cell etc.)

Fig. 1 caption: “redbrown” - not sure what that means...

We have altered this to the more standard “red-brown”, but could also opt for a different name if editorial steer suggests a better one.

Fig. 1 structures are a bit too busy in our opinion -- should be clipped more tightly (true of some other figures too, like Fig 3D-E).

We have trialled modified figures with stronger clipping but the outcome of this lost detail when comparing the structures with one another; for reasons of cross-referencing between the figures we would like to keep the clipping minimal.

Fig. 1 could show rotation arrows to indicate which view angle is which

Agreed. This information has been included on a new version of Figure 1.

Fig. 2: add titles to panels so don't have to look back at caption all the time

The titles have been added to each panel.

Fig. 2 caption (& S5): how much of each effector (uM)? “Saturating amounts”? Or at least “see Methods”.

A note to “See materials and methods for concentrations of effectors used” has been added.

A/B/C etc. panel labels are quite small (too small?) in some figures.

As per our response to reviewer 1, we can subdivide figures up if deemed necessary.

Fig. 5B bottom (sequence diagram) could be moved up to Fig. 1A, before structures. As-is, we had to look forward to Fig. 5B several times before it was even introduced.

We apologise that this does not come sooner, but argue that it is best placed here such that it can be used to interpret the full-length structures in the accompanying panels, in the first figure that these are introduced in – it would uncouple this relationship to shift this to figure 1 (where it has less context next to an ME-only structure).

Fig S1C: void = what? Unfolded due to mutation?

The void volume of the column used as a signifier of aggregated protein – yes, as a result of mutation influencing stability. We have added the statement “A peak in the void region is suggestive

of the E544R mutation negatively influencing protein stability.” to the figure legend.

Fig. S6 should say what is being monitored on the y-axis. We had to dig into the Methods to see that this wavelength (apparently) monitors acetyl-CoA. In general, could include a bit more in the captions about what we’re seeing -- or at least redirect to Methods in key spots.

This information has been added to the caption. “HPLC-UV was used to monitor the change in absorbance at λ_{max} for CoA (259 nm).”

Figure S7: Should show where the green vs. blue parts are in the structure overlay!

This has now been altered in a new version of figure S8 (was S7 before addition of additional acetyl contacts figure).

Table S1: X-ray stats table: Need to say what #s in () mean.

The existing table legend (line 762 of original submission) provides this information as “Numbers in parentheses refer to the outermost shell” (as is standard).

Table S1 does not list the PDB IDs, despite the facts that they are listed elsewhere in the paper and that Table S1 has an empty row for them...

We apologise for this omission and have added this information to a new Table S1.

Table S3 (crystallization conditions) seems to have an error in merging the first two rows, leading to initial confusion about which condition goes with which structure.

This has now been corrected in a new Table S3.

Line 53 typo

Was fixed in response to reviewer 1.

Line 122 typo

Was fixed in response to reviewer 1.

Finally, PLEASE PUT FIGURES AND CAPTIONS IN-LINE NEAR THE RELEVANT PORTIONS OF THE TEXT! Or at the very least, PUT THE FIGURE CAPTIONS NEXT TO THE FIGURES! It makes the reviewers’ jobs much harder when we have to juggle 3-4 different PDFs and Word docs on-screen at once... If this formatting is due to journal policy rather than author choice, PLEASE CHANGE THE POLICY!

We apologise for this situation and will reformat in future.

We review non-anonymously,

Daniel Keedy

Virgil Woods

References

1. Hsieh, J.Y., et al., Structural characteristics of the nonallosteric human cytosolic malic enzyme. *Biochim Biophys Acta*, 2014. 1844(10): p. 1773-83.
2. Hsieh, J.Y., et al., Fumarate analogs act as allosteric inhibitors of the human mitochondrial NAD(P)⁺-dependent malic enzyme. *PLoS One*, 2014. 9(6): p. e98385.
3. Herencias , C., Salgado-Briegas , S., Prieto , M.A., Nogales ,J. Providing new insights on the biphasic lifestyle of the predatory bacterium *Bdellovibrio bacteriovorus* through genome-scale metabolic modelling. *PLoS Comput Biol* 2020 Sep 14;16(9):e1007646. doi: 10.1371/journal.pcbi.1007646. eCollection 2020 Sep

REVIEWER COMMENTS

Reviewer #1 (Remarks to the Author):

The authors have made some changes in the text and added one figure to address my comments.

It is unfortunate that they appear not to have made any changes in the other structure figures, as I still think that the quality of these figures is poor.

The added figure of the acetyl group binding site (together with Fig. S2) shows that few of the side chains contacting this group is conserved, calling into question the generality of their observations with this enzyme (which is supported by their kinetic data showing that the Bd-MaeB is regulated differently compared to *E. coli* MaeB). This should be discussed more carefully in the text.

The authors indicate this is a binding pocket for the acetyl group, and a surface figure (colored electrostatically and/or by conservation) would be helpful (because I fail to see a pocket in the new Fig. S7).

Reviewer #2 (Remarks to the Author):

The authors have answered all the raised concerns.
I think this manuscript represents a worthy contribution and merits publication.

Reviewer #3 (Remarks to the Author):

We thank the authors for responding to our comments. For the most part, they have satisfactorily addressed our concerns.

The authors did not provide an updated validation report for the 3.7A structure that addresses the errors therein. (This is a related but separate point from the purported interpretability of the map in their hands, which they focus on in their rebuttal.) However, we anticipate that the PDB will provide a corrected validation report upon release of the structures, thus solving this issue for future users of the structures.

The authors did helpfully respond to our question regarding the mechanism of inhibition based on their kinetics results (which Reviewer #1 also found confusing). In their response they say, "The K_i value was only reported for the fixed [NADP⁺] experiment where a non-competitive model is appropriate and graphical plots (Figure S5.I) clearly show a decrease in V_{max} and an unchanged K_M ." We do not think it is abundantly obvious from Figure S5.I that the K_M is unchanged -- this would be more clear if the lines were extended to negative x-axis values (left of the y-axis) to see the x-intercepts. By eye they look similar, but it is a bit tough to be sure.

We disagree that "It is difficult to "show" a feature that loses density/definition" -- this can be shown with a side-by-side 2-panel figure of the electron density maps at identical contour levels for the same region of the structure. However, this is not a key point for these revisions.

All of this said, we do not feel that any of these lingering minor issues should prevent publication of the current revised manuscript. Very nice paper!

We review non-anonymously,
Daniel Keedy

Virgil Woods

We thank the reviews for their time and comments, and our point-by-point responses are detailed below in red.

REVIEWER COMMENTS

Reviewer #1 (Remarks to the Author):

The authors have made some changes in the text and added one figure to address my comments.

It is unfortunate that they appear not to have made any changes in the other structure figures, as I still think that the quality of these figures is poor.

The added figure of the acetyl group binding site (together with Fig. S2) shows that few of the side chains contacting this group is conserved, calling into question the generality of their observations with this enzyme (which is supported by their kinetic data showing that the Bd-MaeB is regulated differently compared to *E. coli* MaeB). This should be discussed more carefully in the text.

The authors indicate this is a binding pocket for the acetyl group, and a surface figure (colored electrostatically and/or by conservation) would be helpful (because I fail to see a pocket in the new Fig. S7).

We agree that we can add to the interpretation of this pocket and so have added two new subpanels to figure S7, where the PTA fold and acetate-binding pocket are coloured by conservation (we used ~5600 sequences and the program CONSURF to perform this calculation). This analysis reveals the mosaic nature of the pocket, formed by residues demonstrating 60-100% conservation; perhaps to be expected given that allosteric features are less than absolutely conserved *c.f.* catalytic features. The CONSURF method/details now form part of an amended figure legend. This pocket (and channel) can be seen more clearly in fig S7C.

For the purposes of discussing variation in regulation in other species more effectively in the text, we have added this “closing” statement to the paragraph in which we reference the known variation of the *S. meliloti* DME, TME, and *A. brasilense* enzymes: “The means by which the allosteric PTA domain communicates to the ME enzyme via use of linker helix rearrangement and contact to the hook region is likely to be common to all MaeB enzymes, but the precise accommodation of allosteric effector at the binding pocket may be tuneable through amino acid variation in differing homologues.” This now forms 15 lines of text in the discussion dedicated to variation, which we hope will guide the reader well in interpretation of our work in relation to other proteins.

Reviewer #2 (Remarks to the Author):

The authors have answered all the raised concerns.

I think this manuscript represents a worthy contribution and merits publication.

We would like to note our thanks.

Reviewer #3 (Remarks to the Author):

We thank the authors for responding to our comments. For the most part, they have satisfactorily

addressed our concerns.

The authors did not provide an updated validation report for the 3.7A structure that addresses the errors therein. (This is a related but separate point from the purported interpretability of the map in their hands, which they focus on in their rebuttal.) However, we anticipate that the PDB will provide a corrected validation report upon release of the structures, thus solving this issue for future users of the structures.

We include an updated validation report that notes the relative quality of our structure to those at the same resolution (demonstrating that our structure is generally as good as or better than these with respect to refinement statistics). To answer the reviewer question about error, we trialed a different refinement procedure in PDB-Redo, and whilst this reduced the R/R_{free} gap by $\sim 2\%$, rotamer quality was affected; hence, we did not pursue this further (this also didn't effect our interpretation of the structure or mechanisms inferred).

The authors did helpfully respond to our question regarding the mechanism of inhibition based on their kinetics results (which Reviewer #1 also found confusing). In their response they say, "The K_i value was only reported for the fixed [NADP+] experiment where a non-competitive model is appropriate and graphical plots (Figure S5.I) clearly show a decrease in V_{max} and an unchanged K_M ." We do not think it is abundantly obvious from Figure S5.I that the K_M is unchanged -- this would be more clear if the lines were extended to negative x-axis values (left of the y-axis) to see the x-intercepts. By eye they look similar, but it is a bit tough to be sure.

We are pleased our explanation of the kinetics results helped with the understanding of the mechanism of inhibition. We have now expanded the x and y axis of figure S5.F,H,J through to negative values. To our eye, Figure S5.J shows clear differences between y-axis intercepts, whereas x-axis intercepts are more convergent upon one point.

We disagree that "It is difficult to "show" a feature that loses density/definition" -- this can be shown with a side-by-side 2-panel figure of the electron density maps at identical contour levels for the same region of the structure. However, this is not a key point for these revisions.

We agree that this is easy enough to show, and so have revised figure S8 (that already centred on hook domain features) to include an additional two panels (D, E) demonstrating this concept side-by-side.

All of this said, we do not feel that any of these lingering minor issues should prevent publication of the current revised manuscript. Very nice paper!

Again, we would like to note our thanks for this careful and considered appraisal of our work.